# Mapping the planet's critical areas for biodiversity and nature's contributions to people

Rachel A. Neugarten [1,2,3] ✉, Rebecca Chaplin-Kramer[4,5], Richard P. Sharp[4,6], Richard Schuster [7,8], Matthew Strimas-Mackey [3], Patrick R. Roehrdanz [2], Mark Mulligan[9], Arnout van Soesbergen [9,10], David Hole [2], Christina M. Kennedy [11], James R. Oakleaf[12], Justin A. Johnson[13,14], Joseph Kiesecker[12], Stephen Polasky[13,14], Jeffrey O. Hanson [8] & Amanda D. Rodewald [1,3]

Meeting global commitments to conservation, climate, and sustainable development requires consideration of synergies and tradeoffs among targets. We evaluate the spatial congruence of ecosystems providing globally high levels of nature's contributions to people, biodiversity, and areas with high development potential across several sectors. We find that conserving approximately half of global land area through protection or sustainable management could provide 90% of the current levels of ten of nature's contributions to people and meet minimum representation targets for 26,709 terrestrial vertebrate species. This finding supports recent commitments by national governments under the Global Biodiversity Framework to conserve at least 30% of global lands and waters, and proposals to conserve half of the Earth. More than one-third of areas required for conserving nature's contributions to people and species are also highly suitable for agriculture, renewable energy, oil and gas, mining, or urban expansion. This indicates potential conflicts among conservation, climate and development goals.

Rapid transformation of ecosystems combined with anthropogenic climate change are driving global declines in biodiversity and nature's contributions to people (NCP)[1]. While habitat conversion for economic development has raised standards of living for many, the loss of NCP has negatively impacted millions of people[2]. In an attempt to prevent further losses of biodiversity and NCP, nearly 200 nations have recently committed to effectively conserving and managing 30% of lands and waters by 2030 under the Kunming-Montreal Global Biodiversity Framework[3] and similar national targets (e.g., "America the Beautiful"[4]). Other proposals call for conserving 50% of global land area, or "Half-Earth", to safeguard biodiversity and avert the most devastating effects of climate change[5,6]. National governments have

[1]Department of Natural Resources and Environment, Cornell University, 226 Mann Drive, Ithaca, NY 14853, USA. [2]Conservation International, 2100 Crystal Drive #600, Arlington, VA 22202, USA. [3]Cornell Lab of Ornithology, Cornell University, 159 Sapsucker Woods Rd, Ithaca, NY 14850, USA. [4]Global Science, WWF, 131 Steuart St, San Francisco, CA 94105, USA. [5]Institute on the Environment, University of Minnesota, 1954 Buford Ave, St. Paul, MN 55108, USA. [6]SPRING, 5455 Shafter Ave, Oakland, CA 94618, USA. [7]Nature Conservancy of Canada, 245 Eglinton Ave East, Suite 410, Toronto, ON M4P 3J1, Canada. [8]Department of Biology, Carleton University, Ottawa, ON, Canada. [9]Department of Geography, King's College London, Bush House, North East Wing, 40 Aldwych, London WC2B 4BG, UK. [10]UN Environment Programme World Conservation Monitoring Centre, 219 Huntingdon Road, Cambridge CB3 0DL, UK. [11]Global Science, The Nature Conservancy, Fort Collins, CO 80524, USA. [12]Global Protect Oceans, Lands and Waters Program, The Nature Conservancy, Fort Collins, CO 80524, USA. [13]Department of Applied Economics, University of Minnesota, St. Paul, MN 55108, USA. [14]Natural Capital Project, University of Minnesota, St. Paul, MN 55108, USA. ✉e-mail: ran63@cornell.edu

also initiated efforts to achieve targets related to the Paris Climate Agreement and the UN Sustainable Development Goals. There is a growing recognition that biodiversity, climate, and development goals are intertwined: addressing climate change is necessary to avoid further losses of biodiversity[1], nature-based solutions are essential for mitigating and adapting to climate change[7], and conserving natural assets and addressing climate are both foundational for achieving sustainable development[8]. At the same time, ensuring food, energy, and livelihood security would require carefully planned development of agriculture, energy, urban expansion, and other sectors[9]. Unfortunately, this kind of coordinated planning is rare; where development is not designed to optimize NCP and biodiversity, conservation and development goals will conflict.

Building on recent work[10], we present maps of joint priorities for ten important NCP as well as terrestrial biodiversity. We include one NCP with global benefits, due to its importance for mitigating climate change: vulnerable terrestrial ecosystem carbon storage, defined as the proportion of total ecosystem carbon that could be lost in a typical disturbance event[11]. We also include nine NCP with local or regional benefits[10]: coastal risk reduction, flood regulation, sediment retention (important for reducing erosion and improving water quality), nitrogen retention for water quality regulation, crop pollination, fodder production for livestock (including grazing), fuel wood production, timber production, and access to nature (important for recreation as well as physical and mental well-being). All NCP are realized, either as an end use or benefit (e.g., timber harvest or livestock fodder production), or, where possible given current data, weighted by number of beneficiaries (e.g., count of people downstream, or count of people with reduced risk from coastal storm surge). We define areas providing 90% of all ten NCP as critical natural assets[10].

As a measure of biodiversity, we use Area of Habitat (AOH) for each of 26,709 terrestrial vertebrate species[12]. We conducted a global spatial optimization at a spatial resolution of 10 km to identify prioritized areas that simultaneously achieve target levels of all ten NCP and also achieve minimum species representation targets, independent of protection status. Species targets were consistent between all scenarios and were intended to represent both restricted-range and wide-ranging species. Following previous studies[12–15] we set representation targets that require 10–100% of each species' AOH be conserved, based on the total AOH area (see "Methods"). We did not set area constraints for each country. To identify potential areas of conflict between conservation and development objectives, we overlaid the prioritized areas with estimates of development potential across major sectors, including commercial agriculture, renewable energy, mining, oil and gas, and urban expansion[16]. While past research has explored the sufficiency of the global protected area (PA) network for biodiversity[13], the sufficiency of the PA network for jointly representing many NCP and species has not been explored. We therefore calculated the percentage of prioritized areas that fall within PAs and other effective area-based conservation measure (OECM) areas. We also ran a separate prioritization analysis to identify areas required to achieve targets beyond the current system of protected areas and OECM, by locking in such areas to the prioritization results.

## Results

### Prioritized areas for NCP and species

Our results indicate that conserving 44% of global land area, excluding Antarctica, could provide 90% of current levels of ten NCP and meet minimum representation targets for 26,709 terrestrial vertebrate species, if spatially optimized and coordinated among nations (Figs. 1 and 2). If the current network of PA and OECM sites are locked in to the optimization results, the percent of global land area required to achieve targets increases to 49%. If species targets are not included, 90% of NCP could be provided with 36% of global land area, but many of the areas required to achieve NCP and species targets overlap

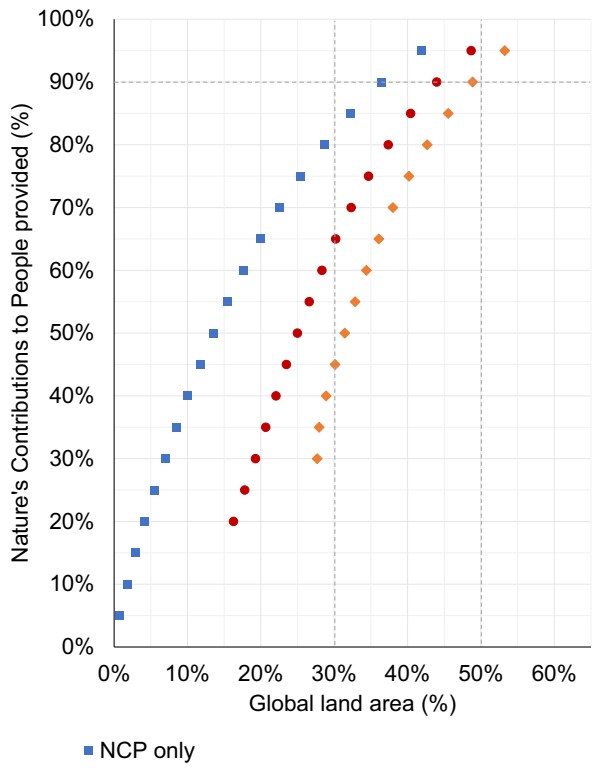

**Fig. 1 | Percentage of global land area required to provide different levels of NCP.** Prioritized areas for nature's contributions to people (NCP) (blue squares) with species targets included (red circles) and with protected areas (PAs) and other effective area-based mechanisms (OECM) sites locked in (orange diamonds). Vertical dashed lines correspond to 30 and 50% of global land area. The horizontal dashed line corresponds to 90% of NCP.

(Fig. 3), demonstrating the opportunity for synergies between maintaining NCP and conserving biodiversity.

Prioritized areas for NCP and species are distributed unevenly across countries, including species-rich areas within the Amazon and Congo basins, Papua New Guinea and Indonesia, and southeastern Australia, which also contain high levels of vulnerable ecosystem carbon storage[11] (Fig. 2a). Other notable areas include regions with unusually high endemism or restricted-range species, including the Himalayas (also important for water quality, flood regulation, and fuelwood production), the Andes (grazing and water quality), New Zealand (water quality, grazing), eastern Madagascar (vulnerable carbon, sediment retention, fuelwood), the Caribbean islands (water quality, pollination, nature access, and grazing), montane regions of Central America (nature access, fuelwood, grazing, pollination), western India (multiple NCP), and islands in Oceania. Western Europe (nature access, pollination and grazing) and the Yangtze basin (water quality, flood regulation, pollination, fuelwood, and timber) also provide globally high levels of NCP and important habitat for many species.

While species targets can be achieved with relatively little additional land area, when compared to areas prioritized solely for NCP, the inclusion of species targets changes the spatial distribution of priority areas slightly (Fig. 3). For example, sparsely vegetated arid lands (e.g., southwestern USA, western Australia) and northern latitudes (e.g., northern Canada and Russia), contain important biodiversity but relatively lower levels of the NCP modeled here due to sparse vegetation and/or lower human population densities (Supplementary Fig. 1). The NCP included in this analysis, which can be

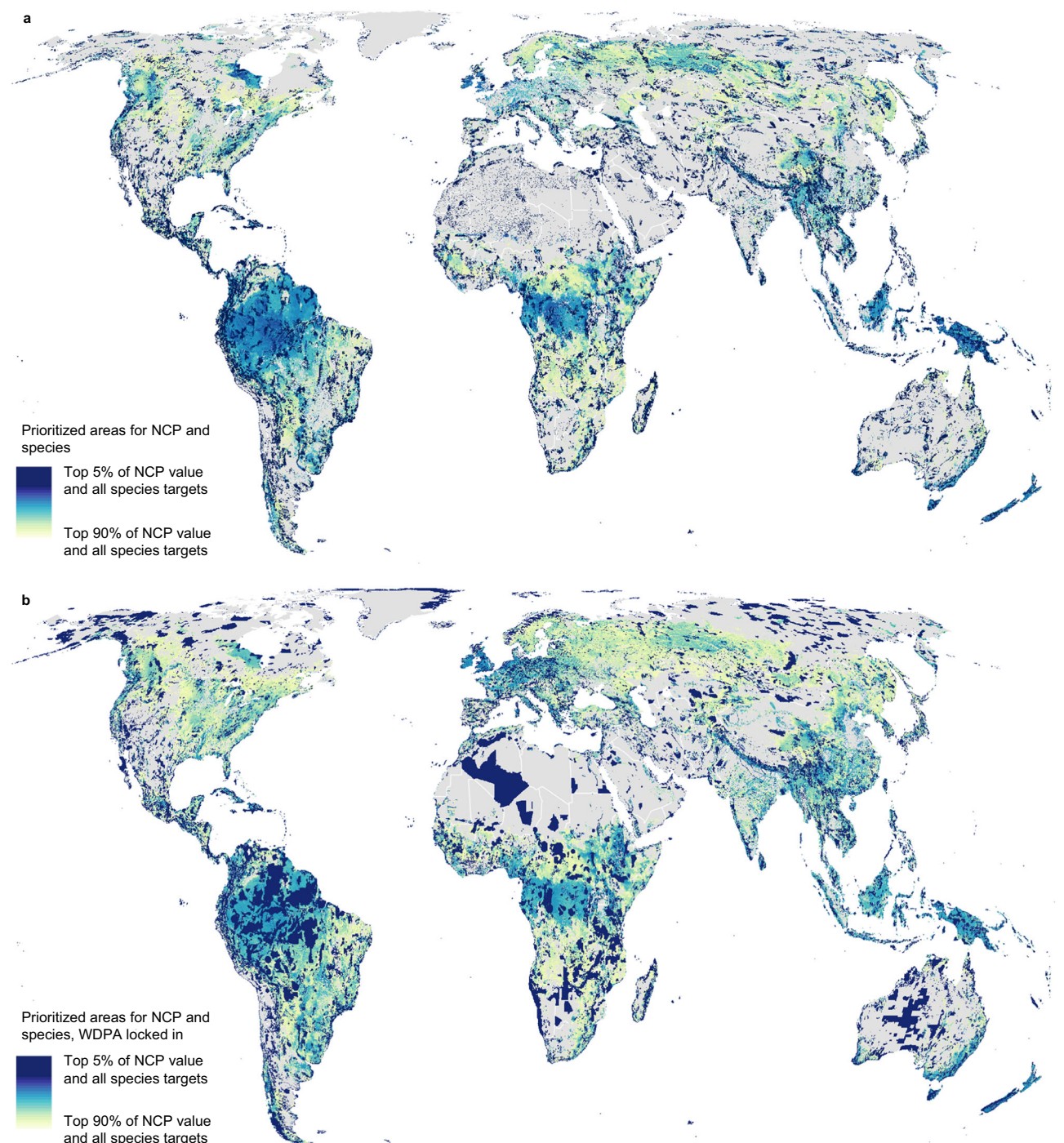

**Fig. 2 | Prioritized areas for nature's contributions to people (NCP) and biodiversity. a** Combined prioritization results for all species representation targets and NCP targets ranging from 5% (dark blue) to 90% (light yellow). **b** Combined prioritization results for NCP and species with the World Database of Protected Areas (WDPA) and other effective area-based conservation mechanisms (OECM) sites locked in to prioritization results. In all cases, dark blue areas represent areas required to achieve targets in the least amount of area. Collectively, dark blue to light yellow areas provide 90% of all ten NCP and meet species representation targets in the least amount of area. Prioritized areas achieve all species representation targets (see main text); only the level of NCP achieved varies.

modeled with globally available data, tend to be concentrated in regions with dense vegetation and in areas accessible to, or upstream of, human populations[10].

Only 18% of the prioritized areas for NCP and biodiversity are currently protected, based on the World Database on Protected Areas (WDPA), which includes other effective area-based conservation measures (OECM)[17] (Supplementary Fig. 2). We found that conserving or sustainably managing an additional 34% of land area beyond the current system of protected areas and OECM (49% of global land area) would be required to provide 90% of current levels of NCP and meet species representation targets (Fig. 1).

## Prioritized areas with high development potential
More than one-third (37%) of areas prioritized for NCP and species also have high development potential for commercial agriculture, renewable energy, oil and gas, mining or urban expansion (equivalent to 16%

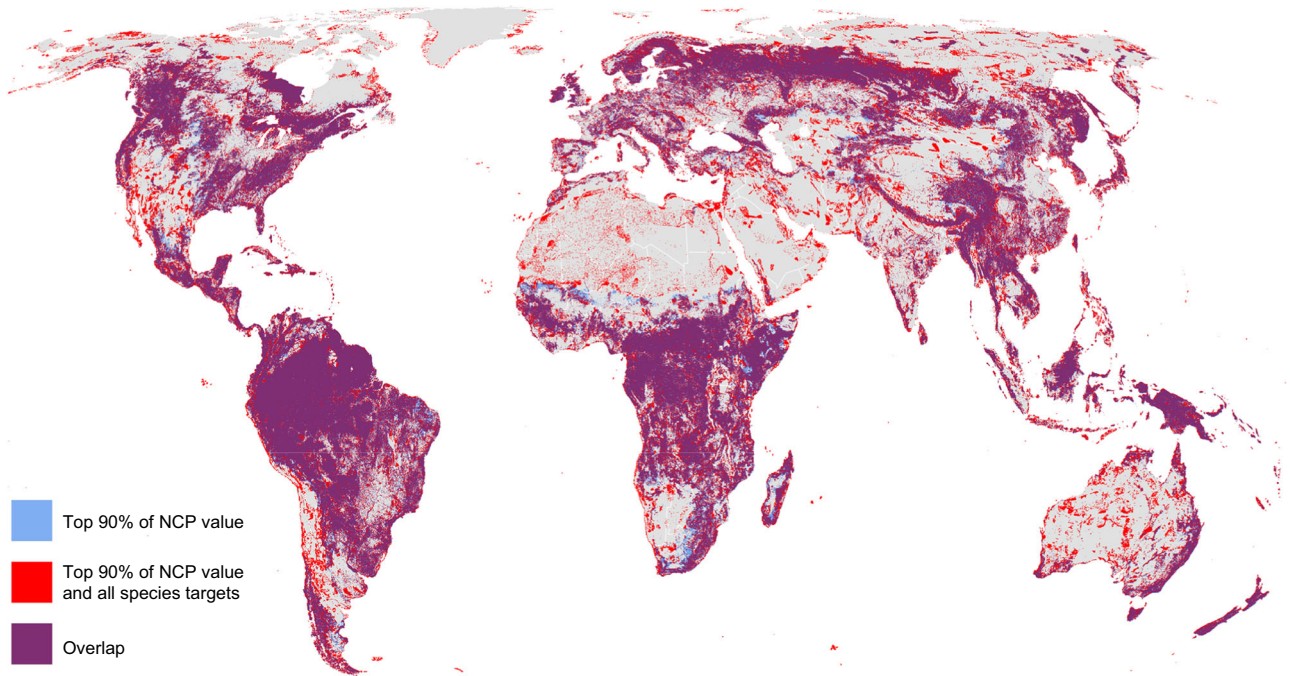

**Fig. 3 | Prioritized areas for nature's contributions to people (NCP) only (90% of current levels of NCP, in blue), prioritized areas for both NCP (90% of current levels) and that also meet all species targets (red), and areas of overlap** **(purple).** Prioritized areas overlap over 33% of global land area (representing 94% of areas prioritized for NCP alone, or 75% of areas prioritized for NCP and species).

of global land area) (Fig. 4a). Only 11% of such areas are currently protected, which may result in future conflicts between development and conservation objectives. The renewable energy sector (concentrated solar power, photovoltaic solar, wind, and hydropower) comprises the largest share of areas with high development potential globally[16], and overlaps with 10% of prioritized areas (4% of global land area) (Fig. 4b). Though renewable energy is needed to avert catastrophic effects of climate change and can be implemented in ways that are compatible with NCP[18] our findings underscore the need to carefully plan, site, and evaluate tradeoffs with other objectives[19,20]. Constraining new projects to already cleared or degraded lands, for example, would reduce conflicts between renewable energy and biodiversity conservation goals[18,20,21].

Areas with high suitability for commercial agriculture (including crops and biofuels) overlap with 7% of prioritized areas (3% of global land area) (Fig. 4b). While agricultural expansion can support food security, if not implemented sustainably, conversion of natural ecosystems to croplands may undermine nature's other contributions[22]. These include benefits to existing agricultural systems such as pollination, sediment retention, and flood mitigation. Policies promoting food security should therefore consider the contributions of croplands as well as natural and semi-natural habitats to food systems.

Mining, which overlaps with 6% of prioritized areas (3% of global land area), and oil and gas development (5% of prioritized areas, 2% of global land area) could create more localized but severe hazards for NCP and species, and are a cause for concern in parts of Western Asia, North America, and the Amazon.

For six of the world's fourteen biomes (broad habitat types), at least one-quarter of their areas contain prioritized areas and are highly suitable for development, making these habitats of special concern. These include mangroves, temperate broadleaf and mixed forests, flooded grasslands and savannas, tropical and subtropical dry broadleaf forests, temperate conifer forests, and temperate grasslands, savannas and shrublands (Supplementary Figs. 3 and 4 and Supplementary Data 1 and 2). In these habitats, future development should be

carefully sited and planned to avoid negatively impacting nature's contributions for people and biodiversity.

Geographically, prioritized areas overlap with areas of high development potential across 31% of the land area of Oceania, 25% of South America, 23% of Europe, 20% of North America, 17% of Africa, 15% of Australia, and 11% of Asia (Supplementary Data 2). More than half of the land area of certain countries such as Gambia (63%), Ireland (60%), and Jamaica (53%) contain globally prioritized areas with high development potential (Supplementary Fig. 4 and Supplementary Data 2). These patterns are driven by the co-occurrence of NCP, species, and development pressures. For example, areas with dense vegetation (such as tropical forests) in proximity to, or upstream of, human populations may be simultaneously important for biodiversity, NCP (carbon storage, provision of timber and fuelwood, access for recreation), while also being highly suitable for certain kinds of development, such as palm oil.

The co-occurrence of NCP, biodiversity and development pressures aren't limited to forests, however. New Zealand, for example, contains large numbers of endemic species as well as extensive areas important for grazing, pollination, and sediment retention, all overlapping with areas of high potential for expansion by oil and gas, mining, and renewable energy. European countries such as Ireland, the UK, Estonia, Latvia, Finland, and the Netherlands contain extensive areas important for grazing, access to nature, and sediment and nitrogen retention that overlap with areas with high development potential for expansion by agriculture, renewable energy, and mining. Conversely, some countries have only a small fraction of their land area in globally prioritized areas suitable for development, including high income countries such as Denmark (1%), Saudi Arabia (4%), and Iceland (4%).

## Discussion
Our study offers a starting point to identify global targets and broad priority regions for conservation and sustainable use investments. We build on a history of efforts to define global biodiversity hotspots[23] by adding two important new considerations: the diverse contributions of

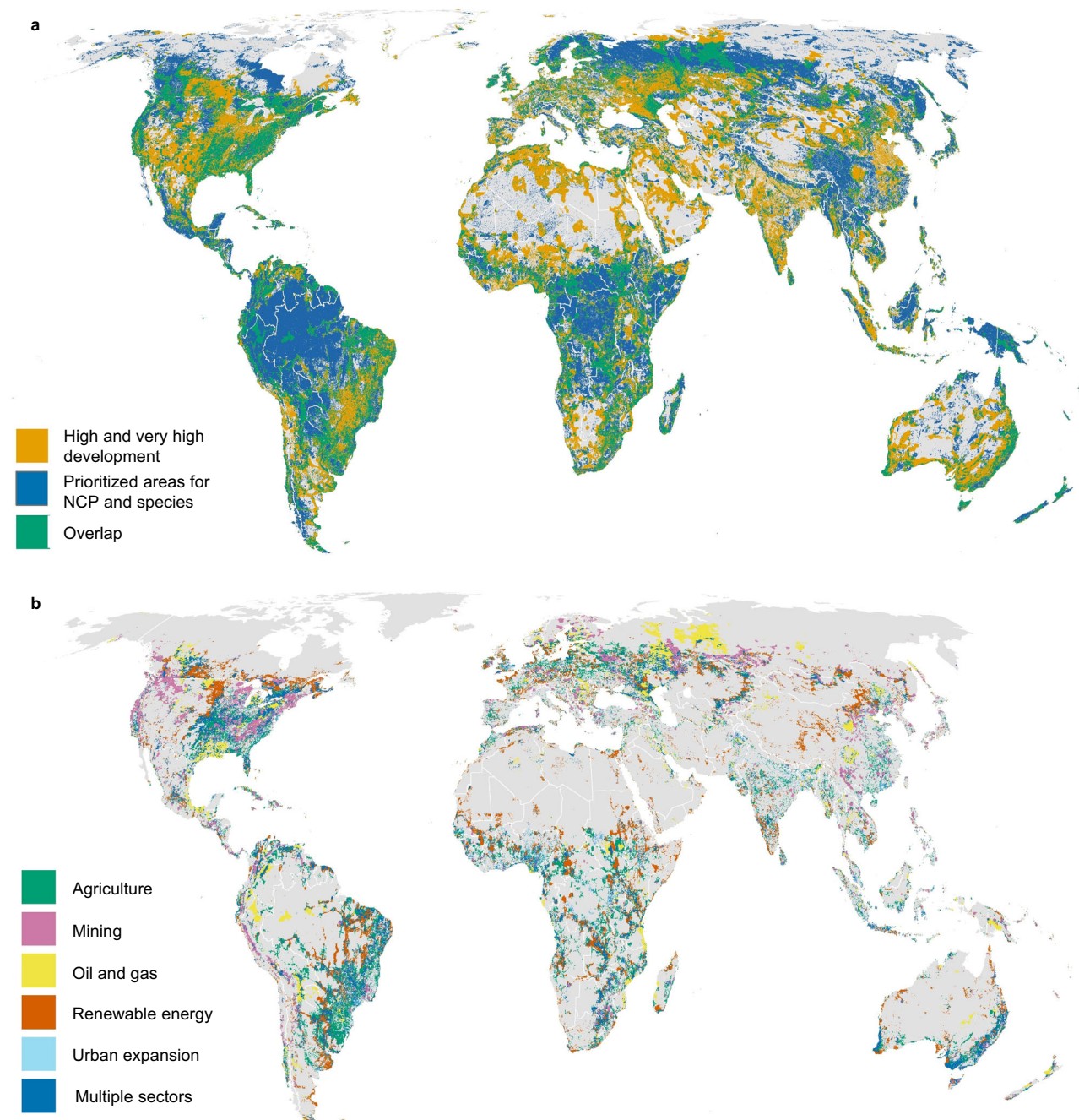

**Fig. 4 | Prioritized areas for nature's contributions to people and biodiversity (NCP) that also have high development potential across several economic sectors. a** Prioritized areas (blue) represent areas providing 90% of current levels of NCP while also achieving all species targets. Areas with high and very high development potential (orange) and areas of overlap (green). **b** Prioritized areas with high development potential (areas of overlap), by economic sector. Sectors include agriculture (crops and biofuels expansion) (green); mining (metallic, non-metallic, and coal) (pink); oil and gas (conventional and unconventional) (yellow); renewable energy (concentrated solar power, photovoltaic solar, wind, and hydropower) (red); urban expansion (light blue), and "multiple sectors" where sectors overlap (dark blue).

nature to people and potential conflicts with expansion of agriculture, energy, extractive industries, or urban development projects. To our knowledge, this is the most comprehensive effort to bring together data on NCP, biodiversity, and development pressures at a global scale. Our results lend support to proposals to conserve at least 30% of the planet by 2030, as well as proposals to conserve "Half-Earth" for biodiversity and its benefits to humanity. Our estimates of the proportion of global land area needed to achieve species and NCP targets (44–49%) are likely conservative, given that estimates based on biodiversity conservation alone range from 34–44%[13,24]. The logistical

challenges of conserving or sustainably managing small non-contiguous priority areas, and the likelihood that not all areas would be effectively conserved, also imply that more land area would be needed to achieve species and NCP targets.

By providing consistent and comparable data across countries, global maps can facilitate the establishment of international targets and highlight where broad action and investment may be most impactful[25]. Our work builds on previous efforts that focused on national-scale NCP priorities that accrue at local- to regional-scales[10]. Here we also include areas required for conserving vulnerable

ecosystem carbon stocks (a global benefit) as well as biodiversity. Our aim in this paper was to identify global-scale priorities to support processes such as the Global Biodiversity Framework, and to inform funding priorities of actors with a worldwide remit. Within priority areas, national and sub-national planning also can benefit from understanding the global significance of local conservation efforts. Global priorities also can support efforts of less wealthy nations to secure resources to achieve shared global targets.

Many NCP (such as water quality regulation, flood mitigation, and carbon storage) cross national borders, as do many species (migratory birds, wide-ranging mammals) therefore identifying areas of global importance is a key first step. Nonetheless, we recognize that most development and conservation decision-making takes place at national and sub-national scales. In previous work, we provided national-scale priorities for NCP[10]. Here, we provide globally optimized results disaggregated by country (Supplementary Figs. 3 and 4 and Supplementary Data 1 and 2). We also provide spatial results from the global optimization scenarios to support finer-scale prioritization or decision making[26]. Other recent work has compared global and national-scale priorities for biodiversity and carbon[27]. In all cases, finer-scale information related to conservation feasibility, costs, and the rights and preferences of local people should be combined with global- or national-scale priorities to identify appropriate interventions for particular locations[25]. Furthermore, conservation and development projects should always be co-developed in partnership with Indigenous peoples and local communities to respect local perspectives and sovereignty, and to result in more effective and equitable outcomes[28].

While our maps identify areas in urgent need of conservation attention, they are not intended to define priorities for strict protection. Strictly protected areas preclude activities such as grazing or timber harvesting which are essential to the provision of certain NCP. Furthermore, the current PA and OECM networks are disproportionately located in remote areas with relatively low threat[29], and do not represent important areas for NCP particularly well (Supplementary Fig. 2), as NCP tend to be concentrated in areas with natural and semi-natural habitat in proximity to human populations. That said, due to data limitations for both NCP and biodiversity, we do not recommend degazetting currently protected areas on the basis of our maps alone. Other conservation measures, including OECMs, strengthening Indigenous and local land tenure, Payments for Ecosystem Services (PES), and sustainable management will be essential for conserving NCP and biodiversity outside of the current system of protected areas. For example, areas providing high levels of water quality, flood regulation, and timber production could be targeted for PES, certification, or other mechanisms. Areas required to achieve species targets that also contain vulnerable carbon could be candidates for Indigenous, local, or government protection; but methods other than protection can also be effective at maintaining biodiversity and carbon stocks. Our maps of prioritized areas include both natural and semi-natural (e.g., grazed pasture, commercial forestry) landcover classes. In such areas, the goal would be to maintain sustainable flows of NCP while also conserving biodiversity.

Our optimization results represent a best-case scenario in which conservation efforts are internationally coordinated. Our findings indicate that conserving 30% of global land area could, if optimally allocated, represent areas supplying 65% of current levels of NCP while also meeting species representation targets. If the current system of PAs and OECM are locked in to the optimization scenario, 30% of land area only provides 45% of NCP while also achieving species targets. This provides a clue that expansion of protected areas, even if nations were to reach a 30% area target, will at best represent 45% of current levels of NCP. Also, given the many barriers to optimally targeting conservation action and investments, our estimates of the area required to achieve targets are likely conservative.

Conversely, areas not identified as priorities in our analysis may contain valuable NCP and biodiversity that are not well represented in globally available data, therefore supplementing global-scale priorities with local data, as well as data on NCP and biodiversity not represented here, is essential[25]. Furthermore, our maps provide an indicator of areas where certain land uses may conflict with conservation in the future, but shifts in demand for energy and commodities, and ever-changing policies and incentives, make it very challenging to predict exactly which areas will actually be developed[19]. Our global estimates of areas with high suitability generally reflect patterns of expansion when production demands are considered[19], and capture areas of projected tree cover loss[30] and urban and cropland expansion[31] by other studies (Supplementary Fig. 5). However, we recognize that our development pressure map may under- or over-estimate development threats in certain regions. Where possible, new development should be constrained to already cleared or degraded areas[18-20]. Certain forms of development, if appropriately located and carefully designed, may be compatible with the ongoing provision of NCP and biodiversity conservation. Examples include water-sensitive urban design that enhances biodiversity, such as green roofs and rain gardens[32] and solar energy farms that can double as livestock enclosures[33], enhance crop production[34] or provide habitat for pollinators and other ecosystem services[35].

The large disparities between countries in terms of levels of NCP, biodiversity, and development potential highlights the importance of international cooperation[27]. Countries with larger conservation responsibilities but without sufficient domestic resources will require access to international funding from their wealthier peers, via mechanisms such as the Global Environmental Facility[36]. The Global Biodiversity Framework includes a target of increasing biodiversity-related funding from developed countries to developing countries to at least USD 30 billion per year by 2030[3]. An alternative approach is for all countries to set consistent targets, such as conserving thirty percent of their land area[4]. While resulting in more equitable distribution of land areas between countries, consistent area-based targets requires more land area overall to achieve targets[10,27], and risks missing the mark for NCP and species which are disproportionately concentrated in a minority of countries.

While our analysis includes a large number of NCP and species, the areas we identified are still an underestimate of the true extent of natural ecosystems needed to sustain all life on earth. Advances in NCP modeling and data availability will soon make it possible to model additional NCP at global scales[37]. Biodiversity priorities will benefit from data on additional taxa such as plants[38] and invertebrates, marine[39] and freshwater[40] species. More comprehensive biodiversity priorities should also incorporate other important dimensions of biodiversity such as evolutionary processes, species traits[41], intactness of ecosystems[42], and ecosystem representation[43], many of which are represented in Key Biodiversity Areas[44]. Computational limitations constrained the spatial resolution of our analysis; future research to conduct prioritizations using more powerful computing resources could advance our understanding of finer-scale patterns and priorities. Projected future demand for NCP due to changes in climate, population, and consumption patterns, as well as species responses to climate change, could help identify ecosystems that will become critical in the future[45].

To date, international agreements such as the recently adopted Global Biodiversity Framework and the Paris Climate Agreement have largely ignored nature's many other contributions to human life and well-being. The natural ecosystems we identify here underpin at least half of the UN Sustainable Development Goals, including providing clean water, reducing hunger, contributing to climate resilience, renewable energy, and supporting health and well-being[8]. Given that every person on the planet benefits from nature, and ~87% of the global population, 6.4 billion people, benefit locally from critical natural

assets[10], conservation and climate targets should more explicitly incorporate the many other benefits that nature provides to humanity. Future global negotiations must move beyond considering biodiversity, climate, and sustainable development in isolation to jointly consider the multitude of nature's contributions to life, livelihoods, and cultures on earth.

## Methods

### Nature's contributions to people

We included global maps of ten NCP that were first mapped in a previous analysis[10] in a new optimization framework which includes biodiversity. The NCP included here are vulnerable ecosystem carbon storage, coastal risk reduction, flood regulation, sediment retention, nitrogen retention, crop pollination, fodder production for livestock (including grazing), fuel wood production, timber production, and access to nature. Vulnerable ecosystem carbon storage is mapped as the above-ground and below-ground ecosystem carbon lost in a "typical" disturbance event[11]. This includes terrestrial and coastal (mangrove, salt marsh, seagrass) ecosystem carbon pools (above-ground, below-ground, and soils), based on how much carbon is likely to be released if the ecosystem were converted. Coastal protection, sediment retention, nitrogen retention, and crop pollination were modeled using InVEST models[46], adapted to be run at global scales[10,45]. Fodder production for livestock, timber production, and fuelwood production were modeled using Version 3 of Co$ting Nature[10,47]. Flood regulation was modeled using Version 2 of WaterWorld[10,48,49]. Access to nature was modeled as the number of urban and rural people[50] within one hour of travel of natural and semi-natural habitat, taking the least-cost path (by foot, road, rail or boat) across a friction surface developed using data on roads, railroads, rivers, bodies of water, elevation and slope, land cover, and national borders[10,51]. This layer may overestimate the accessibility of nature for people who don't have access to cars, and it doesn't account for access rights nor physical barriers such as fences. Data sources, units of measurement, and the original spatial resolution of each modeled NCP are summarized in Supplementary Table 1; additional datasets included in our analysis are summarized in Supplementary Table 2.

All NCP are realized, either as an end use or benefit (e.g., timber harvest or livestock fodder production per unit area of land), or, where possible given current data, weighted by number of beneficiaries[10]. Beneficiaries include people downstream of habitats providing flood regulation or water quality benefits, coastal populations protected from coastal storm surge, or people within a certain travel time of natural habitats. We attributed all NCP to the natural and semi-natural land cover classes providing the benefit, excluding developed lands (croplands and urban areas) and unvegetated areas (Supplementary Table 3). We excluded Antarctica due to lack of data on NCP from that continent.

### Biodiversity (area of habitat, AOH)

As a measure of biodiversity, we used species area of habitat (AOH)[12] for all 26,709 species of birds, mammals, reptiles and amphibians for which data was available. AOH are based on species range maps from IUCN but refined using habitat preferences and elevational limits from IUCN Red List data[12]. AOH are more specific than extent-of-occurrence (EOO) which can overestimate species range sizes[52]. AOH areas "exclude areas of unsuitable habitat from each species' range, which reduces commission errors and more closely approximates the actual occurrence of the species"[53].

Species AOH ranges were produced for all terrestrial vertebrates for which IUCN range polygon data is available[12]. This includes 10,774 species of birds, 5219 mammals, 4462 reptiles and 6254 amphibians. Species range polygons obtained from the IUCN Red List spatial data portal[54] and the Birdlife International spatial data zone[55] were first filtered for "extant" range then rasterized to a global one km

grid in the Eckert IV equal area projection. Individual species range rasters were then modified to only include land cover classes that match the habitat associations for each species. Habitat associations were obtained from the IUCN Red List species habitat classification scheme and were matched to ESA land cover classes for the year 2018[56]. ESA land cover classification data was aggregated from its native 300 m resolution to match the global ten km grid using a majority rule. Species ranges were additionally filtered so that only areas within a species' accepted elevational range were included. Global elevation data derived from SRTM was obtained from WorldClim v. 2[57]. For bird species, seasonal range codes 1–3 (1 = year-round; 2 = breeding range; 3 = non-breeding range) were processed individually and stored as separate range files where applicable. Species targets used for the spatial optimization are described below.

### Spatial optimization

We used linear programming techniques[58] to estimate how much land area is needed to provide different levels of NCP and/or achieve species representation targets. We identified areas that provide the highest value across all ten NCP using spatial prioritization procedures. Specifically, we generated prioritizations using the minimum set formulation of the reserve selection problem, and completing optimization procedures using linear programming techniques[15]. These procedures were completed using the prioritizr R package[15,59] and Gurobi[60]. Because our global-scale optimization included a large number of planning units (more than 20 million) along with 26,709 species and ten NCP features, prioritizr stood out as both computationally efficient and, when combined with Gurobi, sufficiently powerful to solve large optimization problems[59].

The minimum set formulation of the reserve selection problem seeks to minimize the overall cost of the prioritization, whilst ensuring that representation targets are met for all of the conservation features. To define this formulation mathematically, let $I$ denote the set of planning units (indexed by $i$) and $J$ denote the set of features (indexed by $j$). Also, let $c_i$ denote the cost of planning units $i \in I$, $r_{ij}$ denote the amount of features $j \in J$ in planning units $i \in I$, and $T_j$ denote the targets for feature $j \in J$. Additionally, the decision variables are the $x_i$ variables, which indicate if planning units $i \in I$ are selected, or not, for prioritization (using values of one and zero, respectively). Given these variables, the problem can be formulated following:

$$\text{Minimize} \sum_{i=1}^{I} x_i c_i \qquad (1)$$

subject to

$$\sum_{i=1}^{I} x_i r_{ij} \geq T_j \forall j \in J \qquad (2)$$

The objective function (Eq. (1)) is to minimize the cost of selected planning units. Constraints (Eq. (2)) are used to ensure that the representation targets are met.

To explore the land area required to maintain different levels of NCP provision, we ran the optimization using 19 different targets ranging from 5 to 95% of total NCP value, across all ten NCP, at 5% increments.

To explore how much additional area is required to conserve biodiversity, we added species representation targets, using data on extent of suitable habitat, or area of habitat (AOH)[12] data for all species of mammals, reptiles, amphibians, and birds for which data was available (26,709 species in total). We followed previous studies which established targets based on species' habitat size, with the goal of ensuring that both restricted-range and wide-ranging species are represented[12–15]. We assigned a 100% threshold to species with less than 1000 km² of suitable habitat (2391 species of amphibians, 1024

birds, 680 mammals, and 1264 reptiles), a 10% threshold to species with more than 250,000 km² of suitable habitat (695 amphibians, 5600 birds, 1758 mammals, and 589 reptiles), and log-linearly interpolated thresholds for species with intermediate amounts of suitable habitat (2872 amphibians, 6296 birds, 2607 mammals, and 2168 reptiles; migratory bird species were assigned targets for each seasonal distribution separately). We also assigned a cap of 1,000,000 km² for species with a large amount of suitable habitat (>10,000,000 km²) (six amphibians, 148 birds, 57 mammals, and six reptiles). These targets should be considered minimum representation targets as they do not account for habitat connectivity, ecological intactness[42], species traits[41], evolutionary processes, ecosystem representation[43], genetic diversity, or other important dimensions of biodiversity. Species targets are also summarized in Supplementary Table 4. Species targets were consistent across all scenarios (that is, NCP targets varied from 5–95% across scenarios, but species targets were achieved in all scenarios.)

To develop the maps in Fig. 2, we combined (summed) the optimization results for NCP targets ranging from 5–90%. Darker blue areas in Fig. 2 provide the highest levels of NCP per unit area (collectively providing 5% of current levels of all ten NCP in the least area) and lighter yellow areas provide lower levels of NCP per unit area (collectively, the dark blue to light yellow areas provide 90% of all ten NCP). Species targets were held constant across all scenarios.

For subsequent analyses, we focused on prioritized areas, defined as areas providing 90% of current levels of all ten NCP which also meet all species targets. We overlaid prioritized areas with data on development potential to examine overall conversion risk as well as risks from major sectors.

Separately, we also ran scenarios in which protected areas from the World Database on Protected Areas[17] were locked in to the spatial prioritization (that is, protected areas were required as part of the solution in each scenario). This allowed us to estimate how much additional land area would be required to achieve NCP and species representation targets, beyond the current system of protected areas (Figs. 1 and 2 and Supplementary Table 5).

For the scenarios that included NCP (but not species), we ran the optimizations globally and at a spatial resolution of two km. Due to computational constraints and the large number of planning units (more than 20 million) and species (26,709), 10 km was the finest resolution at which we were able to run optimizations for scenarios that contained both NCP and species targets. After running the optimizations, we masked the 10 km optimization results using natural and semi-natural landcover data[61] at two km. This allowed us to more precisely map and quantify the natural and semi-natural habitats providing NCP, and to align our results with NCP-only scenarios run at a higher spatial resolution[10]. The NCP models used here assume low or no provision of NCP in sparsely vegetated areas (such as the extremely arid deserts of the Sahara, the Australian outback, and the Arabian peninsula) and human-modified habitats (such as intensive croplands and developed urban areas.) Many species rely on deserts and modified landscapes, however. To address this, we re-included the prioritized areas for species (at 10 km), ensuring that species that rely on deserts and human-modified habitats were included.

Computational limitations associated with optimizing across the large number of planning units (more than 20 million) and features (more than 26,000) prevented use of a contiguity criterion, which would have selected adjacent (contiguous) planning units when possible. Consequently, though our results are area efficient, they include prioritized areas that are not contiguous in certain regions and may thus require additional planning for implementation. Global priorities such as those provided here can be informative, but should always be combined with local information, including existing land use, to inform decision making.

In the present study, solutions which achieved all targets in the least amount of land area (minimum land area) are used in place of a minimum cost objective. Globally available data on opportunity costs of conservation for agriculture (e.g., Naidoo and Iwamura[62]) have limitations (e.g., lack of information about potential land uses other than agriculture) and were considered unsuitable for this analysis. Proxy indicators such as gridded GDP were also considered a poor measure of cost since costs of safeguarding NCP and biodiversity may be poorly correlated with national or local estimates of economic productivity. Areas with high value for NCP and biodiversity that also have high suitability for development may have high opportunity costs of conservation, due to their potential value for alternative land uses. Instead of a measure of cost, therefore, we separately include data on development potential across major sectors[16].

### Development pressure

We integrated spatially-explicit estimates of the potential for habitat conversion for development across several economic sectors[16]. To create a development pressure map (Supplementary Fig. 6), we used published Development Potential Indices (DPIs)[16] for renewable energy (concentrated and photovoltaic solar power, wind power, and hydropower), oil and gas (conventional and unconventional oil and gas), mining (coal, metallic and non-metallic mining), and commercial agriculture (crop and biofuels expansion). For urban expansion pressure, we created an Urban Pressure Index (UPI) following similar methodologies and categorization techniques as the DPIs using urban expansion probabilities[63] (see Supplementary Materials for details on the UPI).

DPIs are global, spatially explicit one km resolution maps that depict the suitability of land for potential expansion by agriculture, renewable energy, oil and gas, mining, and urbanization. Each DPI has standardized 0–1 values that account for sector-specific land constraints that restrict development (e.g., suitable land cover, slope); land suitability for sector expansion based on resource availability (sector-specific yields); and siting feasibility of new development (e.g., ability to transport resources or materials, access to demand centers, existing development, and other economic costs associated with resource siting). For each of the 14 DPIs, we binned the range of values represented into six categories based on standardized z-score ranges to characterize development pressure as very low (≤10th percentile), low (>10th–25th percentile), medium-low (>25th–50th percentile), medium-high (>50th–75th percentile), high (>75th–90th percentile), and very high (>90th percentile). We calculated z-scores by mean-standardizing values per country to capture national-level domestic demand coupled with global-level demand likely to drive national-level resource extraction to occur within each countries' highest development suitability for that resource. To identify regions of high development pressure, we retained the highest value within the 14 classified DPIs (Supplementary Fig. 6a) and then selected the high and very high classified cells (i.e., values 5 and 6) (Supplementary Fig. 6b).

### Protected areas

We evaluated the extent to which currently protected areas and other effective area-based conservation measures (OECM) might achieve targets and maintain NCP, and how much additional land area might require conservation or stewardship. For this step, we compiled spatial data to delineate the boundaries of protected areas and other conservation areas worldwide. To achieve this, we obtained the World Database on Protected Areas (WDPA) and the World Database on Other Effective Area-Based Conservation Measures (WDOECM)[17]. We prepared these data for analysis following standard practices (using the wdpar R package)[64]. Briefly, we (1) excluded sites within an unknown or proposed designation, (2) excluded UNESCO Biosphere Reserves[65], (3) transformed the site boundaries to avoid numerical issues associated with geometries that cross the dateline,

(4) reprojected the data to an equal-area coordinate reference system (World Behrmann; ESRI:54017), (5) replaced sites represented as point localities with circular reserves matching their reported area[66], (6) removed slivers and (7) excluded marine protected areas. Additionally, throughout this process, we implemented routines to detect and repair invalid geometries.

We included WDPA and OECM areas in our analysis in two different ways. First, to calculate the percentage of prioritized areas that are currently protected or effectively conserved, we overlaid prioritized areas with the combined WDPA and OECM areas (Supplementary Fig. 2). Second, to calculate how much additional land area would be required to achieve NCP and species representation targets, beyond the current system of protected areas, we ran separate spatial optimization scenarios in which WDPA and OECM areas were locked in (Fig. 2b). Because protected areas and OECM do not necessarily overlap with prioritized areas, locking them in to the prioritization solutions results in larger total land areas to achieve targets (Fig. 1).

### Spatial resolution

To test the effect of spatial resolution on our results, we conducted prioritizations for NCP at four different resolutions: two, three, five, and 10 km. At coarser resolutions, more land area is required to achieve NCP targets (Supplementary Fig. 7 and Supplementary Table 6), consistent with previous studies[67]. Due to the global geographic scope and the large number of species (26,709), prioritizations at finer spatial resolutions for both NCP and species were beyond the scope of this analysis, which relied on traditional computational resources. To address this issue, and to bring our results in line with previous work[10], we masked the 10 km prioritization results to natural and semi-natural habitat data at a finer spatial resolution (two km), which more precisely identifies the two km habitat grid cells which provide NCP within each 10 km grid cell. Spatial analyses other than optimizations were conducted using R[68], QGIS[69], and ArcGIS Desktop[70].

### Reporting summary

Further information on research design is available in the Nature Portfolio Reporting Summary linked to this article.

## Data availability

The data on prioritized areas and high development potential areas generated in this study have been deposited in the Zenodo database[26]. The prioritized areas data disaggregated by country, continent, and biome generated in this study are provided in Supplementary Data files 1 and 2. Data on nature's contributions to people used in this study are available in the Open Science Framework database[71] and can be visualized at: https://bit.ly/3Jk8vDo. The biodiversity data used in this study are available under restricted access for non-commercial use, access can be obtained by request (https://www.iucnredlist.org/resources/spatial-data-download). The WDPA and OECM data used in this study are available under restricted access for non-commercial use, access can be obtained by request (www.protectedplanet.net). The vulnerable carbon data used in this study are available in the Zenodo database[72]. The data on projected tree cover loss used in this study are also available in the Zenodo database[73]. The data on areas vulnerable to land cover change used in this study are available from ArcGIS Online (https://www.arcgis.com/home/item.html?id=645c280931ac486cadb92c828eac09e3).

## Code availability

Code is available on Zenodo: https://zenodo.org/record/8225989.

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

## Acknowledgements

The authors gratefully acknowledge Christopher Barrett, Pamela Collins, Alexandra Goldstein, Catherine Kling, Jeffrey Milder, Monica Noon, and

Will Turner for their insight and intellectual contributions. We gratefully acknowledge funding support from the Cornell University Department of Natural Resources and the Environment (R.A.N.) the Cornell Lab of Ornithology (R.A.N., A.D.R., M.S.M.), Betty and Gordon Moore (R.A.N., P.R.R., D.H.), Conservation International (R.A.N., P.R.R., D.H.), The Natural Capital Project (R.C.K., R.P.S.), SPRING (R.C.K., R.P.S.), John and Jody Arnhold (P.R.R.), Environment and Climate Change Canada (J.O.H.), Nature Conservancy of Canada (J.O.H., R.S.) The Nature Conservancy (C.M.K., J.R.O., J.K.), the Liber Ero Fellowship (R.S.), and One Earth (C.M.K., J.R.O.). We thank the National Science Foundation for funding through Graduate Research Fellowship DGE—2139899 (R.A.N.) and NSF awards 2225078 and 2225076 (P.R.R.). EC Horizon 2020 ReSET project grant 101017857 (M.M., A.v.S.), UKRI/F4B Nature Finance Seed Corn Grant (M.M., A.v.S.).

## Author contributions

R.A.N., R.C.K., D.H., S.P., and A.D.R. conceptualized and designed the analysis. R.A.N., R.C.K., R.P.S., R.S., M.S.M., P.R.R., M.M., A.v.S., C.M.K., J.R.O., J.K., J.A.J., S.P., and J.O.H. contributed to the acquisition, analysis, and interpretation of data. R.P.S., M.M., J.A.J., and J.O.H. created new software and code used in the analysis. R.A.N., R.C.K., and A.D.R. drafted and substantially revised the manuscript. A.D.R. supervised the study. All authors contributed to reviewing and revising the manuscript and approved the final submitted version.

## Competing interests
