## [Peer Review File · Nature Communications]

Mapping the planet's critical areas for biodiversity and nature's contributions to peopleREVIEWER COMMENTS

Reviewer #1 (Remarks to the Author):

This is a very interesting and original paper that extends the concepts of the classic 'hotspots' analysis of Myers to functional ecosystem services. The hotspots approach had major impact on policy-making and funding allocations for biodiversity conservation. One of the flaws of the hotspots approach is that it had narrow species-based criteria that did not include ecosystem services and so excluded many functionally important areas. This paper has the potential for similar policy impact by identifying areas of conflict with alternative land use. Of particular interest is the global scale of the analysis and it was particularly interesting, for example, to see temperate regions such as western UK included as being of high importance. Inevitably, analysis at this scale cannot take into account the local situation, but equally they can stimulate local policy decisions. To come back to the example of priorities in the UK - the analysis will give analytical strength to discussions about 'lost temperate rainforests' in popular media by placing them in a global context.

I consider the methods of analysis to be sound within the limitations discussed above and the interpretations and conclusions to be consistent with the analysis. Sufficient information is given for the analysis can be replicated. The authors mention the SDGs and this could be expanded with a few sentences to place the results in the SDG framework, which is important for guiding national policies. The way that economic activity is implemented has a major effect on levels of biodiversity. It is perfectly possible to include biodiversity considerations in urban expansion through, for example, water sensitive urban design that enhances biodiversity in urban areas and there are many small-scale examples of this. Similarly, renewable energy projects can enhance biodiversity such as off-shore wind farms being de facto marine reserves and solar energy farms can be used to create species-rich meadows. Intensive agriculture is the main threat to terrestrial biodiversity and this can be mitigated using regenerative agriculture techniques that have been proven to be more productive and resilient to climate extremes. Some mention of these possibilities would offer ways forward for sustainable development.

Reviewer #2 (Remarks to the Author):

Overall

This is an interesting paper which contributes to the discussion of how much of the earth to protect, where it should happen, and how that would affect people. I also appreciate the analysis of suitability for development which highlights threats. They draw on appropriate data sources and analytical methods for the questions they pose. I only have two high-level concerns with the paper. First is that centering on a scenario that would mean unprotecting some existing protected areas is misleading. Shifting to centering the paper on the “PAs and OECMs locked in” scenario would make it more clear what action is needed and where/how conservation efforts should be focused globally. The second is that not much development nor conservation are driven by these kinds of global analyses. Both are often opportunistic, and when they are planned it’s usually at the country level. So the addition of a scenario that requires a consistent protection target across countries would be helpful to represent a somewhat more feasible approach (e.g., 30 by 30 is most often interpreted to ask each country to protect 30%). The companion paper “Chaplin-Kramer, R., Neugarten, R. A., Sharp, R. P., Collins, P. M., Polasky, S., Hole, D., Schuster, R., Strimas-Mackey, M., Mulligan, M., Brandon, C., Diaz, S., Fluet-Chouinard, E., Gorenflo, L. J., Johnson, J. A., Kennedy, C. M., Keys, P. W., Longley-Wood, K., McIntyre, P. B., Noon, M., ... Watson, R. A. (2022). Mapping the planet’s critical natural assets. *Nature Ecology & Evolution*. <https://doi.org/10.1038/s41559-022-01934-5>” includes such a scenario and I think this paper needs something similar.

Abstract

The abstract says that conserving 44% of land would provide 90% of NCP plus hit representation targets. But the results section says “We found that conserving or sustainably managing an additional 34% of land area beyond the current system of protected areas and OECM (49% of global land area) would be required to provide 90% of current levels of NCP and meet species representation targets (Fig. S4).” Unless the authors are proposing opening some existing protected areas to development, the 44% number is irrelevant and misleading, as it represents an undesirable and unlikely scenario. The more useful numbers that belong in the abstract (and that the paper should center on) are the need to conserve 34% additional global land area (bringing it to 49% of global land area, so more or less an endorsement of half-earth). Similarly, I’d recommend moving Fig 1 to the supplement and replacing it with a modified version of Fig S4. Since *not* locking in existing PAs and OECMs seems very unlikely, I’d include the orange line from Fig S4 just labeled as “NCP and species”, and a new line labeled “NCP” that means “NCP, PAs and OECMs locked in.” I think the blue and red lines from Fig S4 should be deleted from any figure in the main text, and honestly don’t think they even belong in the supplement unless the authors add text explaining why such a scenario is useful to consider. Fig 2 through 4 (and several figures in the supplementary materials) would also have to be updated to reflect the locking in of PAs and OECMs.

Introduction

The paper talks a lot about minimum representation targets before defining them. The methods provide clarity, but the paper would be stronger if 1) they added a table to the supplementary materials showing

what % of habitat was considered essential at different sizes of AOH and 2) the first use of the term in the introduction had a very brief definition, e.g. “achieve minimum species representation targets, *which require 10-100% of each species’ habitat be conserved depending on the total habitat area.*” While it doesn’t include connectivity or many other factors, this is actually a higher bar than I would have guessed from the term so the clarity up front is important (this likely makes a big difference to the results).

This sentence could use a brief follow up since this kind of optimization and careful planning is so rare in practice: “At the same time, ensuring food, energy, and livelihood security will require carefully planned development of agriculture, energy, urban expansion, and other sectors⁹.” I’d recommend either 1) change “will” to “would”, and end the sentence with “(and this kind of planning is rare)”, OR 2) adding a sentence like “In a more common scenario where development is *not* optimized for NCP and habitat, more habitat would need to be cleared to meet human needs.” Or something similar. I see the value in these kinds of global analyses, but it’s very important to recognize that they represent a best-case scenario which will not materialize (unless something radically changes), and should not be taken as feasible in the current political climate. Much of global development is not planned by central governments at all (especially ag expansion), and when it is, nature is not often weighted very highly.

One more point is important to convey in the introduction which is currently missing (and the authors make it in related publications). Not only would development need to be planned *within* each country that has regulatory authority (coordinating w/ each other to optimize), we would *also* need to accept that the results would not be equitable across countries. By optimizing on total NCP, some communities and countries will receive more NCP and others will receive less. Similarly, some communities and countries will face more impediments to economic development than others because more of their land will be protected. The supplement mentions that it’s more efficient to optimize globally, but since most protection happens at a national level this also makes the results less practical to implement than a similar analysis constrained by country level targets. If the authors are unable to add a scenario that forces a similar level of protection within each country (as requested in overall comments above), I think they need a couple of sentences owning the implication of this omission on both feasibility and equity.

Results

I don’t understand this sentence as written: “If prioritized together, slightly more land area (8%) is required to achieve all species targets”. As written, it sounds like you need 52% of land area for some higher bar, but from looking at figure 1 it seems like it’s the reverse. I’d recommend rephrasing the second sentence to something like “Without any constraints on retaining representative habitat for vertebrates, 90% of NCP could be provided with 36% of global land area.” And leave off the bit about synergies etc which you discuss later with more useful context. Although as noted above, instead of being 44% and 36%, it should be 49% (locking in PAs & OECMs) and whatever the NCP-only equivalent is that locks in PAs and OECMs.

Figure 1 would benefit from a second pair of dotted lines at 30% of global land area; given the intense focus on 30x30 this paper should make it apparent what % of NCP & species representation could be provided at 30% (if they were allocated optimally).

I very much appreciate the analysis on threats the authors provide. It would be helpful to add a sentence to make explicit what the paper implies: currently protected areas are disproportionately in areas with relatively low threat (since it's easier to protect places that people don't want to develop). It would be helpful for the authors to note that if the intent of conservation and protection is to reduce habitat and species loss, we should instead *focus* protection on areas with more development potential to increase the chance of the protection having an additional / material impact.

Methods

It appears that there is no constraint on the minimum size for protection, nor to reduce the perimeter:area ratio of the resulting portfolio. Please specify whether or not any such constraints were used. If not, I'd recommend briefly noting that while such a scattered / speckled portfolio is efficient from an area perspective, it would be very challenging to implement (both protection designation and enforcement are often challenges in landscapes with this kind of heterogeneous protection).

The methods mention the 30-year time horizon for UPI, but do not specify the time horizon for the DPis. The cited source (Oakleaf et al., 2019) does not make the time horizon obvious either. The development potential will obviously change over time as prices rise and technology makes recovering fossil fuels and minerals more practical (and ag land more productive) so I'd like the development predictions to all use the same time horizon (30 years works well).

While I don't know of a better global data layer of accessibility than Weiss et al. 2018, it would be helpful to note that their travel time is largely driven by roads and travel by car. In developed contexts those roads are often not safe to walk on. So for example there has been recent work showing that some communities are very close physically to nature but would have to jump fences and cross highways to get there (or to have a car and take a longer route). In places with low car ownership the accessibility data should be interpreted with caution.

Discussion:

I appreciate the note that future analyses could include plants and marine ecosystems, but given that freshwater ecosystems are 1) underprotected 2) disproportionately threatened 3) harder to effectively conserve, a call for future research (and protection goals) to include them would be helpful.

There is great content in the discussion. There is only one thing I'd like to see added. The analyses presented show us the best case scenario of what the minimum land needed is under very optimistic conditions. To better drive the point home that we need to change the priority of what actually gets protected, could the authors add any data (or even rough estimates and supposition) around a *lower* bound of benefit if we protect the wrong places? So for example, if you were to use existing PAs and OECMs as an example and extrapolate how similarly suboptimal places would perform, it would be helpful to hear what % of NCP might be realized at similar % of global area protection goals. So if getting to 49% globally optimized is needed to keep 90% NCP and hit minimum representation targets, what might "business as usual" protection look like? I recognize this could be analytically intensive, and don't think the authors should have to do similar analysis if so. But I do think a few sentences pointing out

what BAU could look like would make the paper more impactful by comparing a best case where the paper's recommendations are real to a more likely scenario. Thus the marginal value of the recommendations would be more clear, and it might catch more attention.

Supplement:

It surprised me how different the land needed was for 2km vs 10km. That is what drove the question about any minimum patch size or other constraint. No need to respond unless the authors want to add a sentence or two of insight about these results in the supplement.

Reviewer #3 (Remarks to the Author):

Dear colleagues,

I have thoroughly read your paper and supplementary material. First of all, I believe the paper and its idea are highly relevant and align well with the scope of the journal.

Additionally, I must acknowledge that the models presented in this work are sufficiently intricate to substantiate these comprehensive findings. With that being said, overall, I firmly believe that the paper deserves the opportunity to be published. Although the final decision rests with the editor, I hope my feedback proves helpful in their evaluation process.

I have a couple of major concerns that should be addressed. Depending on the reasons for these issues, it may be necessary to rerun the models and make changes to some of the input data.

Major comments:

My most significant concern pertains to the "areas of conflict" between conservation and human development. I strongly believe that these areas of conflicts are significantly underrepresented in your maps; however, I am unsure of the underlying reason. I have thoroughly reviewed your supplementary information, but this issue is not adequately addressed there either. For instance, I would like to point out that while you include the potential for agriculture, the global deforestation fronts, particularly those directly linked to agriculture, are noticeably underrepresented in your results. You may find it helpful to refer to the information provided at this link:

https://wwf.panda.org/discover/our_focus/forests_practice/deforestation_fronts_/. When examining your results, it becomes apparent, for example, that the Southeast Asia region is not adequately represented. This is extremely worrying as these areas represent the world's most vulnerable, and potentially those with the highest human modification rates during the near future. This was evident to me, but I fear the same could be happening with the rest of the "uses"

My second major concern pertains to the inclusion of biodiversity in your study. I agree with the use of the AOH approach, and I appreciate the way you performed prioritization for the Natural Conservation Priority (NCP) by implementing a simple conservation planning procedure and exploring different thresholds. However, I have noticed that this procedure was not followed when including biodiversity. Upon reviewing Map S1, it became evident

that biodiversity was represented as a binary raster without taking into account essential aspects such as connectivity. As a result, I believe that the prioritization approach used for biodiversity is overly simplistic and may not accurately identify priority areas for biodiversity conservation. Therefore, I recommend conducting another type of prioritization analysis, such as a zonation analysis. This would allow you to consider crucial factors like species conservation status, connectivity, and degree of endemism (which is already "relatively" considered), among many other essential aspects. It is essential here, to include, besides the degree of endemism, aspects such as the vulnerability of each included species. In addition, it is not clear to me why the pool of included species was selected. It is essential to clarify this point.

I have a few other concerns, but I acknowledge that they are minor compared to the ones mentioned earlier. Additionally, considering the extent and resolution of your analyses, it may be challenging to address these concerns effectively. I have intentionally focused on raising concerns that I believe could be helpful in improving your paper. However, I suggest considering the inclusion of a couple of sentences to acknowledge some of the most significant weaknesses of your analysis regarding the input data, extent, and resolution.

I really hope his assessment and point of view could be beneficial in enhancing your paper. Please feel free to contact me if you have any further questions or require clarification. Best regards,

Javier Nori

Reviewer #1 (Remarks to the Author):

This is a very interesting and original paper that extends the concepts of the classic 'hotspots' analysis of Myers to functional ecosystem services. The hotspots approach had major impact on policy-making and funding allocations for biodiversity conservation. One of the flaws of the hotspots approach is that it had narrow species-based criteria that did not include ecosystem services and so excluded many functionally important areas. This paper has the potential for similar policy impact by identifying areas of conflict with alternative land use. Of particular interest is the global scale of the analysis and it was particularly interesting, for example, to see temperate regions such as western UK included as being of high importance. Inevitably, analysis at this scale cannot take into account the local situation, but equally they can stimulate local policy decisions. To come back to the example of priorities in the UK - the analysis will give analytical strength to discussions about 'lost temperate rainforests' in popular media by placing them in a global context.

- Response: We thank the reviewer for their kind comments and are pleased they believe this paper has potential for policy impact. We agree the results highlight regions of global importance, such as temperate systems, which are typically overlooked in international policy settings despite their importance for both biodiversity and people.

Reviewer: I consider the methods of analysis to be sound within the limitations discussed above and the interpretations and conclusions to be consistent with the analysis. Sufficient information is given for the analysis can be replicated. The authors mention the SDGs and this could be expanded with a few sentences to place the results in the SDG framework, which is important for guiding national policies. The way that economic activity is implemented has a major effect on levels of biodiversity. It is perfectly possible to include biodiversity considerations in urban expansion through, for example, water sensitive urban design that enhances biodiversity in urban areas and there are many small-scale examples of this. Similarly, renewable energy projects can enhance biodiversity such as off-shore wind farms being de facto marine reserves and solar energy farms can be used to create species-rich meadows. Intensive agriculture is the main threat to terrestrial biodiversity and this can be mitigated using regenerative agriculture techniques that have been proven to be more productive and resilient to climate extremes. Some mention of these possibilities would offer ways forward for sustainable development.

- Response: Thank you for both these comments. We have expanded our discussion of how our results fit in the SDG framework in the Discussion: “The natural ecosystems we identify here underpin at least half of the UN Sustainable Development Goals, including providing clean water, reducing hunger, contributing to climate resilience, renewable energy, and supporting health and well-being (Hole et al., 2022).”
- We fully agree with the reviewer regarding opportunities to design land uses that protect biodiversity and NCP while accommodating development. We have added examples suggested by the reviewer (as well as a couple more of our own) in the Discussion: “Certain forms of development, if appropriately located and carefully designed, may be compatible with the ongoing provision of NCP and biodiversity conservation. Examples include water-sensitive urban design that enhances biodiversity, such as green roofs and

rain gardens³² and solar energy farms that can double as livestock enclosures³³, enhance crop production³⁴ or provide habitat for pollinators and other ecosystem services³⁵.”

Reviewer #2 (Remarks to the Author)

Overall

This is an interesting paper which contributes to the discussion of how much of the earth to protect, where it should happen, and how that would affect people. I also appreciate the analysis of suitability for development which highlights threats. They draw on appropriate data sources and analytical methods for the questions they pose. I only have two high-level concerns with the paper. First is that centering on a scenario that would mean unprotecting some existing protected areas is misleading. Shifting to centering the paper on the “PAs and OECMs locked in” scenario would make it more clear what action is needed and where/how conservation efforts should be focused globally.

- Response: Thank you for your comments, we endeavored to draw on (or develop) appropriate data and analytical methods to support our analysis. Thank you also for pointing out that our results could be interpreted as implying that some existing protected areas should be un-protected, this was not our intention. We now include both scenarios – one that “locks in” PAs and OECMs and one that doesn’t lock in PAs and OECMs. We retained the scenario that does not lock in PAs and OECMs for several reasons. First, the designation of a site as a protected area does not necessarily indicate the site is effectively conserved in practice (e.g. “paper parks”). Second, because strictly protected areas usually prohibit human activities (such as grazing, fuelwood collection, timber harvesting) which are essential for providing certain NCP, we wanted to avoid implying that prioritized areas should necessarily become strict protected areas. The appropriate conservation approach (protection, sustainable use, or other measures) will depend on a range of socio-political, economic, and cultural considerations which should complement global-scale priorities. We have attempted to explain this more fully in the Discussion, for example: “While our maps identify areas in urgent need of conservation attention, they are not intended to define priorities for strict protection. Strictly protected areas preclude activities such as grazing or timber harvesting which are essential to the provision of certain NCP.”
- Third, we wanted to ask, as an empirical question, the extent to which current PA and OECM sites are located in the most important areas for NCP and species, and which additional areas might require conservation attention to achieve targets. While past research has explored the sufficiency of the global PA network for biodiversity (e.g. Hanson, Rhodes, et al., 2020), to our knowledge, no one has explored the sufficiency of the PA network for jointly representing NCP and species. Answering the first question requires including a scenario in which PA and OECM sites are not locked in. We have now included results from both scenarios in the main text and figures as we recognize that different readers might be interested in different scenarios.

Reviewer: The second is that not much development nor conservation are driven by these kinds of global analyses. Both are often opportunistic, and when they are planned it's usually at the country level. So the addition of a scenario that requires a consistent protection target across countries would be helpful to represent a somewhat more feasible approach (e.g., 30 by 30 is most often interpreted to ask each country to protect 30%). The companion paper “Chaplin-Kramer, R., Neugarten, R. A., Sharp, R. P., Collins, P. M., Polasky, S., Hole, D., Schuster, R., Strimas-Mackey, M., Mulligan, M., Brandon, C., Diaz, S., Fluet-Chouinard, E., Gorenflo, L. J., Johnson, J. A., Kennedy, C. M., Keys, P. W., Longley-Wood, K., McIntyre, P. B., Noon, M., ... Watson, R. A. (2022). Mapping the planet's critical natural assets. *Nature Ecology & Evolution*. <https://doi.org/10.1038/s41559-022-01934-5>” includes such a scenario and I think this paper needs something similar.

- Response: We agree with the reviewer's point about the importance of country-level information for practitioners and decision-makers and are interested in doing this in the future. As the reviewer notes, in previous work, we provided national-scale priorities for NCP. The focus of the current paper was defining global-scale priorities to support international processes such as the UN Global Biodiversity Framework. Because many species cross national borders, we felt that global-scale optimizations were more appropriate for this initial step. Also, defining appropriate national-scale targets for species is challenging, particularly for wide-ranging species that span multiple countries, therefore we felt doing so was not justified for the current paper. We do provide the global results disaggregated by country in Supplementary Tables 7 and 8, to inform national-scale priorities, and we make our spatial results available for download.
- We have added text to try to explain our rationale more carefully in the Discussion:
- “By providing consistent and comparable data across countries, global maps can support the establishment of international targets and highlight where broad action may be most impactful. Our aim in this paper was to identify global priorities to support processes such as the Global Biodiversity Framework. We build on previous work which focused on NCP that benefit people on local- to regional-scales by including areas required for conserving vulnerable ecosystem carbon stocks as well as biodiversity. Many NCP (such as water quality regulation, flood mitigation, and carbon storage) cross national borders, as do many species (migratory birds, wide-ranging mammals) therefore identifying areas of global importance is a key first step. Nonetheless, we recognize that most development and conservation decision-making takes place at national and sub-national scales. In previous work, we provided national-scale priorities for NCP. Here, we provide globally optimized results disaggregated by country (Supplementary Figs. 5-6, Supplementary Tables 7-8). We also provide spatial results from the global optimization scenarios to support finer-scale prioritization or decision making (<https://zenodo.org/record/7803242>). Other recent work has compared global and national-scale priorities for biodiversity and carbon. In all cases, finer-scale information related to conservation feasibility, costs, and the rights and preferences of local people should be combined with global- or national-scale priorities to identify appropriate interventions for particular locations. Furthermore, conservation and development projects should always be developed in partnership with Indigenous peoples and local communities to respect local perspectives and sovereignty, and to result in more effective and equitable outcomes.”

Reviewer: Abstract

The abstract says that conserving 44% of land would provide 90% of NCP plus hit representation targets. But the results section says “We found that conserving or sustainably managing an additional 34% of land area beyond the current system of protected areas and OECM (49% of global land area) would be required to provide 90% of current levels of NCP and meet species representation targets (Fig. S4).” Unless the authors are proposing opening some existing protected areas to development, the 44% number is irrelevant and misleading, as it represents an undesirable and unlikely scenario. The more useful numbers that belong in the abstract (and that the paper should center on) are the need to conserve 34% additional global land area (bringing it to 49% of global land area, so more or less an endorsement of half-earth). Similarly, I’d recommend moving Fig 1 to the supplement and replacing it with a modified version of Fig S4. Since *not* locking in existing PAs and OECMs seems very unlikely, I’d include the orange line from Fig S4 just labeled as “NCP and species”, and a new line labeled “NCP” that means “NCP, PAs and OECMs locked in.” I think the blue and red lines from Fig S4 should be deleted from any figure in the main text, and honestly don’t think they even belong in the supplement unless the authors add text explaining why such a scenario is useful to consider. Fig 2 through 4 (and several figures in the supplementary materials) would also have to be updated to reflect the locking in of PAs and OECMs.

- Response: We again appreciate the reviewer pointing this out and have included both scenarios in the abstract, main text, and figures. We have attempted explain our reasoning for including both scenarios above.
- The revised abstract now reads: “We find that conserving 44-49% of global land area through protection or sustainable management could provide 90% of current levels of ten of nature’s contributions to people and meet minimum representation targets for 26,709 terrestrial vertebrate species. This finding supports recent commitments by national governments under the Global Framework for Biodiversity to conserve at least 30% of global lands and waters, and proposals to conserve “half Earth”.
- Figure 1 now includes results from the NCP-only scenario, the NCP + species scenario, and the NCP+ species with PA and OECMs locked in scenario, and the accompanying has been updated.
- Figure 2 now includes a third map (originally included as Supplemental Fig S3) with the PAs and OECMs locked in scenario, and we have moved the accompanying descriptive text up so it is now highlighted more clearly.

Reviewer: Introduction

The paper talks a lot about minimum representation targets before defining them. The methods provide clarity, but the paper would be stronger if 1) they added a table to the supplementary materials showing what % of habitat was considered essential at different sizes of AOH and 2) the first use of the term in the introduction had a very brief definition, e.g. “achieve minimum species representation targets, *which require 10-100% of each species’ habitat be conserved depending on the total habitat area.*” While it doesn’t include connectivity or many other factors, this is actually a higher bar than I would have guessed from the term so the clarity up front is important (this likely makes a big difference to the results).

- Response: Thank you for these suggestions and clarifications. We have added a table as suggested (Supplementary Table 3), as well as text explanation (very first section of the Supplementary text, “Species targets”). We have also included counts of species falling

into each AOH size category in the main text, Methods section: “We assigned a 100% threshold to species with less than 1,000 km² of suitable habitat (2,391 species of amphibians, 1,024 birds, 680 mammals, and 1,264 reptiles), a 10% threshold to species with more than 250,000 km² of suitable habitat (695 amphibians, 5,600 birds, 1,758 mammals, and 589 reptiles), and log-linearly interpolated thresholds for species with intermediate amounts of suitable habitat (2,872 amphibians, 6,296 birds, 2,607 mammals, and 2,168 reptiles; migratory bird species were assigned targets for each seasonal distribution separately). We also assigned a cap of 1,000,000 km² for species with a large amount of suitable habitat (>10,000,000 km²) (six amphibians, 148 birds, 57 mammals, and six reptiles).”

- In the Introduction we added the suggested text: “Species targets were consistent between all scenarios and were intended to represent both restricted-range and wide-ranging species. Following previous studies^{6,15–17} we set representation targets which require 10-100% of each species’ AOH be conserved, based on the total AOH area (see Methods).”
- In the Methods, we had explained the targets used for species with different sizes of AOH in the section “Spatial optimization”: “We assigned a 100% threshold to species with less than 1,000 km² of suitable habitat, a 10% threshold to species with more than 250,000 km² of suitable habitat, and log-linearly interpolated thresholds for species with intermediate amounts of suitable habitat. We also assigned a cap of 1,000,000 km² for species with a large amount of suitable habitat (>10,000,000 km²).”

Reviewer: This sentence could use a brief follow up since this kind of optimization and careful planning is so rare in practice: “At the same time, ensuring food, energy, and livelihood security will require carefully planned development of agriculture, energy, urban expansion, and other sectors⁹.” I’d recommend either 1) change “will” to “would”, and end the sentence with “(and this kind of planning is rare)”, OR 2) adding a sentence like “In a more common scenario where development is *not* optimized for NCP and habitat, more habitat would need to be cleared to meet human needs.” Or something similar. I see the value in these kinds of global analyses, but it’s very important to recognize that they represent a best-case scenario which will not materialize (unless something radically changes), and should not be taken as feasible in the current political climate. Much of global development is not planned by central governments at all (especially ag expansion), and when it is, nature is not often weighted very highly. One more point is important to convey in the introduction which is currently missing (and the authors make it in related publications). Not only would development need to be planned *within* each country that has regulatory authority (coordinating w/ each other to optimize), we would *also* need to accept that the results would not be equitable across countries. By optimizing on total NCP, some communities and countries will receive more NCP and others will receive less. Similarly, some communities and countries will face more impediments to economic development than others because more of their land will be protected. The supplement mentions that it’s more efficient to optimize globally, but since most protection happens at a national level this also makes the results less practical to implement than a similar analysis constrained by country level targets. If the authors are unable to add a scenario that forces a similar level of protection within each country (as requested in overall comments above), I think they need a couple of sentences owning the implication of this omission on both feasibility and equity.

- Thank you also for these detailed suggestions, in which we have implemented within the Introduction (e.g., “Unfortunately, this kind of planning is rare; where development is not designed to optimize NCP and biodiversity, conservation and development goals will conflict.”)
- Regarding inequities between countries, we also fully agree with this point. We have attempted to explain our focus on global-level priorities for this paper above, but we do recognize the implications of this in terms of different burdens among countries. In the Discussion, we have added further text elaborating on this point: “The large disparities between countries in terms of levels of NCP, biodiversity, and development potential highlights the importance of international cooperation (Shen et al., 2023). More than half of the land area of certain countries such as Gambia (63%), Ireland (60%), and Jamaica (53%) contain globally prioritized areas with high development potential (Supplementary Fig 6, Supplementary Table 7). Conversely, a number of countries have only a small fraction of their land area in globally prioritized areas suitable for development, including high income countries such as Denmark (1%), Saudi Arabia (4%), and Iceland (4%). Countries with larger conservation responsibilities but without sufficient domestic resources will require access to international funding from their wealthier peers, via mechanisms such as the Global Environmental Facility (Xu et al., 2021). The Global Biodiversity Framework includes a target of increasing biodiversity related funding from developed countries to developing countries to at least USD 30 billion per year by 2030 (UN CBD, 2022). An alternative approach is for all countries to set consistent targets, such as conserving thirty percent of their land area (U.S. Department of Interior (DOI), 2021). While resulting in more equitable distribution of land areas between countries, consistent area-based targets requires more land area overall to achieve targets (Chaplin-Kramer et al., 2022; Shen et al., 2023), and risks missing the mark for NCP and species which are disproportionately concentrated in a minority of countries”

Reviewer: Results

I don't understand this sentence as written: “If prioritized together, slightly more land area (8%) is required to achieve all species targets”. As written, it sounds like you need 52% of land area for some higher bar, but from looking at figure 1 it seems like it's the reverse. I'd recommend rephrasing the second sentence to something like “Without any constraints on retaining representative habitat for vertebrates, 90% of NCP could be provided with 36% of global land area.” And leave off the bit about synergies etc which you discuss later with more useful context. Although as noted above, instead of being 44% and 36%, it should be 49% (locking in PAs & OECMs) and whatever the NCP-only equivalent is that locks in PAs and OECMs.

- Response: Thank you for this suggestion, we have revised the text:
- “Our results indicate that conserving 44% of global land area, excluding Antarctica could provide 90% of current levels of ten NCP and meet minimum representation targets for 26,709 terrestrial vertebrate species, if spatially optimized and coordinated among nations (Fig. 1). If the current network of PA and OECM sites are “locked in” to the optimization results, the percent of global land area required to achieve targets increases to 49%. If species targets are not included, 90% of NCP could be provided with 36% of global land area, but many of the areas required to achieve NCP and species targets overlap (Fig 3), demonstrating the opportunity for synergies between maintaining NCP and conserving biodiversity.”

Reviewer: Figure 1 would benefit from a second pair of dotted lines at 30% of global land area; given the intense focus on 30x30 this paper should make it apparent what % of NCP & species representation could be provided at 30% (if they were allocated optimally).

- Response: Thank you, we have added dotted lines at 30% (and also 50%, representing half earth), as well as adding the PA and OECM locked in scenario as suggested above.

Reviewer: I very much appreciate the analysis on threats the authors provide. It would be helpful to add a sentence to make explicit what the paper implies: currently protected areas are disproportionately in areas with relatively low threat (since it's easier to protect places that people don't want to develop). It would be helpful for the authors to note that if the intent of conservation and protection is to reduce habitat and species loss, we should instead *focus* protection on areas with more development potential to increase the chance of the protection having an additional / material impact.

- Response: Thank you for this suggestion. We have added the following text to the Discussion: "The current PA network is disproportionately located in remote areas with relatively low threat (Joppa & Pfaff, 2009), and as our results indicate, does not represent important areas for NCP particularly well. Our maps are intended to fill this gap by identifying places where biodiversity, NCP, and development pressures overlap."

Reviewer: Methods

It appears that there is no constraint on the minimum size for protection, nor to reduce the perimeter:area ratio of the resulting portfolio. Please specify whether or not any such constraints were used. If not, I'd recommend briefly noting that while such a scattered / speckled portfolio is efficient from an area perspective, it would be very challenging to implement (both protection designation and enforcement are often challenges in landscapes with this kind of heterogeneous protection).

- Response: This is correct. We wanted to understand how much land area is required to achieve NCP and species targets, and to understand whether current proposed targets such as 30x30 are sufficient, which was why we didn't include an area constraint. Due to computational limitations we were unable to include a perimeter:area ratio constraint, due to the size of the optimization problem (large number of species features and global extent).
- We agree with the reviewer that where this results in a scattered/speckled portfolio, it introduces challenges from an implementation perspective.
- We have clarified this in the Methods section ("Spatial optimization"): "We wanted to understand how much land area is required to achieve targets, and to understand whether current targets such as "30 by 30" are sufficient, therefore we did not impose a minimum area constraint. Computational limitations associated with optimizing across the large number of planning units (more than 20 million) and features (more than 26,000) prevented use of a contiguity criterion, which would have selected adjacent (contiguous) planning units whenever possible. Consequently, though our results were area efficient, they included prioritized areas that were not contiguous in certain regions and may thus require additional planning for implementation. As stated above, global priorities can be

informative but should always be combined with local information, including existing land use, to inform decision making.”

Reviewer: The methods mention the 30-year time horizon for UPI, but do not specify the time horizon for the DPIs. The cited source (Oakleaf et al., 2019) does not make the time horizon obvious either. The development potential will obviously change over time as prices rise and technology makes recovering fossil fuels and minerals more practical (and ag land more productive) so I'd like the development predictions to all use the same time horizon (30 years works well).

- Response: Thank you for this question, we want to clarify that the DPI models gradient of land suitability and does not model land conversion events within a given time horizon. Doing so requires more detailed information on country-level demand estimates and regional knowledge of governmental actions (e.g., environmental regulations, incentives, tax breaks), market changes and technological advancements. All of these factors are extremely difficult if not impossible to capture spatially accurate and complete data at a global scale. We included this caveat within the Development Pressure section of the Supplementary Materials. However, the high development pressures we mapped do reflect general patterns of expansion when production demands are considered (Johnson et al. 2021), thus, our patterns appear to reflect areas vulnerable to potential expansion by various sectors, although should not be interpreted as the exact location of future development siting.
- While it is accurate that the Zhou et al. (2019) data used to produce the UPI span from 2020 to 2050, we transformed and categorized these data to create a suitability gradient similar to the DPIs. We provide more details on the UPA in the Supplementary Materials. To eliminate the confusion associated with the UPI, we modified the Methods (Development Potential Index section) to read: “For urban expansion pressure, we created an Urban Pressure Index (UPI) following similar methodologies and categorization techniques as the DPIs using future urban expansion probability data (see Supplementary Materials for details on the UPI).”

Reviewer: While I don't know of a better global data layer of accessibility than Weiss et al. 2018, it would be helpful to note that their travel time is largely driven by roads and travel by car. In developed contexts those roads are often not safe to walk on. So for example there has been recent work showing that some communities are very close physically to nature but would have to jump fences and cross highways to get there (or to have a car and take a longer route). In places with low car ownership the accessibility data should be interpreted with caution.

- Response: Weiss et al. 2018 used many datasets beyond roads to create a friction surface: “The datasets that we used to construct the friction surface characterize the spatial locations and properties of roads, railroads, rivers, bodies of water, topographical conditions (elevation and slope angle), land cover, and national borders.” (<https://www.nature.com/articles/nature25181>). We do recognize, however, that the Weiss et al. dataset does allow the fastest mode of transport to take precedence. So we agree with the reviewer that the presence of a road does not necessarily mean people that can use it (especially if they don't have cars) and, likewise, the travel time to any given

natural ecosystem doesn't necessarily indicate accessibility to people (e.g., it may be on private property, behind a fence or other barrier, policed by armed guards, etc.) Effectively, this means that our map of "access to nature" over-estimates the number of people who have physical access to natural and semi-natural ecosystems. We have added a new text to explain this in the Methods section:

- "Access to nature is expressed as the number of urban and rural people (Cattaneo et al., 2021) within one hour travel of natural and semi-natural habitat, taking the least-cost path (by foot, road, rail or boat) across a friction surface developed using data on roads, railroads, rivers, bodies of water, elevation and slope, land cover, and national borders (Weiss et al., 2018). This layer may overestimate the accessibility of nature for people who don't have access to cars, and it doesn't account for physical barriers such as fences. Data sources, units of measurement, and the original spatial resolution of each modeled NCP are summarized in Supplementary Table 1. Full model details, as well as modeling assumptions and limitations, are available in the supplementary materials from (Chaplin-Kramer et al., 2022)"

Reviewer: Discussion:

I appreciate the note that future analyses could include plants and marine ecosystems, but given that freshwater ecosystems are 1) underprotected 2) disproportionately threatened 3) harder to effectively conserve, a call for future research (and protection goals) to include them would be helpful.

- Response: We fully agree, and have added a reference to freshwater biodiversity (and a citation, Tickner et al. 2020):
- "Biodiversity priorities will benefit from data on additional taxa such as plants (Jung et al., 2021) and invertebrates, marine (Sala et al., 2021) and freshwater (Tickner et al., 2020) species, as well as other important dimensions of biodiversity, such as evolutionary processes, species traits (Harris et al., 2021), intactness of ecosystems (Riggio et al., 2020), and ecosystem representation (Sayre et al., 2020).

Reviewer: There is great content in the discussion. There is only one thing I'd like to see added. The analyses presented show us the best case scenario of what the minimum land needed is under very optimistic conditions. To better drive the point home that we need to change the priority of what actually gets protected, could the authors add any data (or even rough estimates and supposition) around a *lower* bound of benefit if we protect the wrong places? So for example, if you were to use existing PAs and OECMs as an example and extrapolate how similarly suboptimal places would perform, it would be helpful to hear what % of NCP might be realized at similar % of global area protection goals. So if getting to 49% globally optimized is needed to keep 90% NCP and hit minimum representation targets, what might "business as usual" protection look like? I recognize this could be analytically intensive, and don't think the authors should have to do similar analysis if so. But I do think a few sentences pointing out what BAU could look like would make the paper more impactful by comparing a best case where the paper's recommendations are real to a more likely scenario. Thus the marginal value of the recommendations would be more clear, and it might catch more attention.

- Response: This is a very interesting question! While we did not analyze this specifically, we believe that some clues exist in our current results, and have added mention of this to the Discussion. As a thought exercise, one could randomly protect pixels and see how much NCP and species are represented. Hypothetically, randomly protecting 30% of the planet should, on average, represent 30% of NCP and 30% of species, without any optimization. However, because the current system of PA and OEEM are not optimally located for NCP and species, the actual number will be lower. We have added the following to the Discussion:
- “Our optimization results represent a best-case scenario in which conservation efforts are internationally coordinated. Our findings indicate that conserving 30% of global land area could, if optimally allocated, represent areas supplying 65% of current levels of NCP while also meeting species representation targets. If the current system of PAs and OEEM are locked in to the optimization scenario, 30% of land area only provides 45% of NCP while also achieving species targets. This provides a clue that expansion of protected areas, even if nations were to reach a 30% area target, will at best represent 45% of current levels of NCP. Other conservation measures, including OEEMs, strengthening Indigenous and local land tenure, and Payments for Ecosystem Services (PES), will be essential for achieving targets outside of the current system of protected areas.”

Reviewer: Supplement:

It surprised me how different the land needed was for 2km vs 10km. That is what drove the question about any minimum patch size or other constraint. No need to respond unless the authors want to add a sentence or two of insight about these results in the supplement.

- Response: We also were surprised by this finding, but were reminded that it is consistent with past research showing that the spatial resolution of the analysis affects the results (e.g. Arponen et al 2012 <https://doi.org/10.1111/j.1523-1739.2011.01814.x>) As the reviewer perhaps noted, we included some explanation about the effect of spatial resolution on our results, and how we addressed them, in the Supplementary Text (section "Spatial resolution"):
- "The spatial resolution of input data, as well as the method used for converting data between spatial resolutions (resampling), is known to influence results of spatial prioritizations. To test the effect of spatial resolution on our results, we conducted prioritizations for NCP at four different resolutions: 2km, 3km, 5km, and 10km. At coarser resolutions, more land area is required to achieve NCP targets (Supplementary Fig. 5, Supplementary Table 5), consistent with previous studies (e.g. Arponen et al. 2012). Due to the large number of species (26,709), prioritizations at finer spatial resolutions for both NCP and species were beyond the scope of this analysis, which relied on traditional computational resources. To address this issue, and to bring our results in line with previous work², we masked the 10 km prioritization results to natural and semi-natural habitat data at a finer spatial resolution (2 km), which more precisely identifies the 2 km habitat grid cells which provide NCP within each 10 km grid cell. "

Reviewer #3 (Remarks to the Author):

Dear colleagues,

I have thoroughly read your paper and supplementary material. First of all, I believe the paper and its idea are highly relevant and align well with the scope of the journal. Additionally, I must acknowledge that the models presented in this work are sufficiently intricate to substantiate these comprehensive findings. With that being said, overall, I firmly believe that the paper deserves the opportunity to be published. Although the final decision rests with the editor, I hope my feedback proves helpful in their evaluation process.

- Response: Thank you for these kind comments, we do very much hope the paper aligns with the scope of Nature Communications and have endeavored to develop or synthesize the best available data and models to support our findings.

Reviewer: I have a couple of major concerns that should be addressed. Depending on the reasons for these issues, it may be necessary to rerun the models and make changes to some of the input data.

Major comments:

My most significant concern pertains to the "areas of conflict" between conservation and human development. I strongly believe that these areas of conflicts are significantly underrepresented in your maps; however, I am unsure of the underlying reason. I have thoroughly reviewed your supplementary information, but this issue is not adequately addressed there either. For instance, I would like to point out that while you include the potential for agriculture, the global deforestation fronts, particularly those directly linked to agriculture, are noticeably underrepresented in your results. You may find it helpful to refer to the information provided at this link:

https://wwf.panda.org/discover/our_focus/forests_practice/deforestation_fronts_/. When examining your results, it becomes apparent, for example, that the Southeast Asia region is not adequately represented. This is extremely worrying as these areas represent the world's most vulnerable, and potentially those with the highest human modification rates during the near future. This was evident to me, but I fear the same could be happening with the rest of the "uses"

- Response: Thank you for this comment and the link. We used the Development Potential Indices because, to our knowledge, these are the most comprehensive global data that consistently maps areas with high suitability for development across multiple development sectors. Although our development pressure map does not predict specific locations that will be converted, it does identify areas that are expected to have greater vulnerability to land conversion or competition that may conflict with safeguarding of NCP or biodiversity (in other words, areas that have globally high values of NCP and/or species as well as high or very high suitability for development). Areas with high development pressure overlap with over a third (37%) of the prioritized areas in our analysis, which we feel does not under-represent areas of potential conflict. We have updated Figure 4 to show the full extent of high development potential areas (previously,

the map only showed prioritized areas that had high development potential (the “intersection”), the updated version shows the full extent of DPI areas overlaid on top of prioritized areas, including high DPI areas that extend beyond prioritized areas (the “union”).

- Nonetheless, we do recognize that certain threats (such as deforestation) or areas (such as SE Asia) may be under-represented in any map that relies on globally available data. We have examined the deforestation fronts website provided by the reviewer. Unfortunately, as far as we can ascertain, the deforestation frontiers data appears to end in 2017, and does not appear to project future deforestation. Our maps of prioritized areas for NCP and biodiversity are based on remaining natural and semi-natural habitat, so areas that have already been deforested will not show up as priorities (and will therefore not represent areas of potential conflict, which is what the reviewer is requesting.)
- To address the reviewer’s concern, we searched for other global-scale products which may provide insights on threats from future deforestation and/or land cover conversion. In order to be compatible with our analysis, such products must be globally comprehensive (which rules out pan-tropical datasets, for example) and be at a spatial resolution that is similar to those used in our analysis (ideally, 10km or finer). Two products met our criteria: global projected tree cover loss to 2029 (Hewson et al., 2019) (<https://zenodo.org/record/3237796>) and projected land conversion (urban and cropland expansion) from 2018 to 2050 (Esri et al., 2021). We note the second product is available for download (<https://www.arcgis.com/home/item.html?id=645c280931ac486cadb92c828eac09e3>) but, to our knowledge, has not been published in the peer-reviewed literature. Also, this product focuses on agriculture and urban expansion, and is thus limited on the sector drivers it considers. Thus we included it here (and in the Supplemental Materials) only for illustrative purposes.
- We have overlaid these two products with our map of prioritized areas and found that, globally, both products have significantly less overlap with prioritized areas than our development pressure map. Areas with “high” and “very high” development pressure overlap with 37% of prioritized areas, while only 15% of such areas are projected to be deforested by 2029, and 19% are projected to have some form of land conversion by 2050. While it’s not possible to directly compare these three products, given their vastly different goals and methodologies, we can conclude that the DPI is a less conservative estimate of areas where conservation and other land use goals may conflict in the future.
- We have included a new figure in the Supplement illustrating these new results (Fig S7).
- Thus, while we recognize that any global map may under-represent threats from specific drivers and/or threats in specific regions, we do believe the DPI serves as a useful indicator of areas where conservation and development objectives may conflict in the future, and overall is less conservative than other global products available that focus on specific habitat types (forest) or specific sectors (i.e., agriculture and urban). We have added the following text to the Discussion:
- “Furthermore, our maps provide an indicator of areas where certain land uses may conflict with conservation in the future, but shifts in demand for energy and commodities, and ever-changing policies and incentives, make it very challenging to predict exactly which areas will actually be developed²⁰. Our global estimates of areas with high suitability generally reflect patterns of expansion when production demands are

considered²⁰, and capture areas of projected tree cover loss³⁰ and urban and cropland expansion³¹ by other studies (Supplementary Fig. 7). However, we recognize that our development pressure map may under- or over-estimate development threats in certain regions. Where possible, new development should be constrained to already cleared or degraded areas^{19–21}.

Reviewer: My second major concern pertains to the inclusion of biodiversity in your study. I agree with the use of the AOH approach, and I appreciate the way you performed prioritization for the Natural Conservation Priority (NCP) by implementing a simple conservation planning procedure and exploring different thresholds. However, I have noticed that this procedure was not followed when including biodiversity. Upon reviewing Map S1, it became evident that biodiversity was represented as a binary raster without taking into account essential aspects such as connectivity. As a result, I believe that the prioritization approach used for biodiversity is overly simplistic and may not accurately identify priority areas for biodiversity conservation. Therefore, I recommend conducting another type of prioritization analysis, such as a zonation analysis. This would allow you to consider crucial factors like species conservation status, connectivity, and degree of endemism (which is already "relatively" considered), among many other essential aspects. It is essential here, to include, besides the degree of endemism, aspects such as the vulnerability of each included species. In addition, it is not clear to me why the pool of included species was selected. It is essential to clarify this point.

- Response: Regarding the pool of included species, we included all species of birds, mammals, reptiles, and amphibians for which IUCN range polygon data was available (26,709 in total). We have clarified this in several places in the text, including in the Methods ("Biodiversity (Area of Habitat, AOH)" section) – "Species AOH ranges were produced for all terrestrial vertebrates for which IUCN range polygon data is available (Brooks et al., 2019). This includes 10,774 species of birds, 5,219 mammals, 4,462 reptiles and 6,254 amphibians."
- We recognize that a wide range of criteria can be used to define biodiversity priorities. In this paper, our goal was to synthesize newly available global datasets on a large number of NCP together with species data for the first time. Thus, our intent was not to define new priorities for biodiversity, as there have been many excellent papers which have defined and mapped global biodiversity priorities using a range of criteria and conservation planning approaches (for example Allan et al., 2019; Brooks et al., 2006; Hanson, Rhodes, et al., 2020; Myers, 2003; Rodrigues et al., 2004). Instead, we hoped to use previously published species targets to identify areas of joint importance for NCP and species, and to understand whether proposed targets such as "30 by 30" are sufficient. In general, our optimization uses a target-setting approach based on each species' total Area of Habitat. This means that species will get a higher proportion of their total AOH represented in the optimization result if they have a smaller total AOH, so it addresses endemism (i.e. how small a species' range is). Since vulnerability and conservation status are often correlated with geographic range size (indeed, a species' range size is a criterion for IUCN Red List status), the target setting approach should address these factors also.

- The reviewer correctly notes that we used different methods for setting targets for NCP and biodiversity. For NCP, we set a range of targets representing 5%-95% of current total global NCP value across all 10 NCP and focused subsequent analyses on areas representing 90% of NCP (following Chaplin-Kramer et al. 2022, which found that beyond 90% of NCP, additional units of land area provide relatively little additional NCP value). For species, we used targets that have been used in similar global-scale prioritizations in the past (e.g. Hanson, Rhodes, et al., 2020; Rodrigues et al., 2004; Schuster et al., 2019). Although the optimization analyses output binary values to indicate which places (planning units) were selected to achieve a given set of targets, the results were difficult to visualize on a global scale given the high resolution of the planning units. As such, the optimization outputs generated using the various NCP targets (5%-90%) were summed together to produce a continuous metric to facilitate visualization. Fig 2 shows the summed optimization outputs so the resulting maps appear to be continuous. Results from each individual optimization scenario are binary (0/1) and can be downloaded from: <https://zenodo.org/record/7803242>). We have revised the caption of Fig 2 to clarify what each map represents.
- Because our global-scale optimization included a large number of planning units (> 20 million) along with >26,000 species and NCP features, prioritizr stood out as both computationally efficient and, when combined with Gurobi, sufficiently powerful to solve large optimization problems (Hanson, Schuster, et al., 2020). Regarding implementing a different kind of optimization, such as Zonation, we would welcome such an approach for future work but it was beyond the scope of the current paper.
- For the specific criteria mentioned, such as species conservation status, our optimization results represent all species (including both threatened and non-threatened species) and achieves minimum representation targets for all species (specific targets range from 100% to 10% of each species Area of Habitat, depending on the AOH size). Many species are declining across many taxonomic groups, regardless of threat status (e.g. Díaz et al., 2019; Rosenberg et al., 2019) and therefore we felt it was important to ensure that all terrestrial vertebrate species were represented in our optimization results, including both threatened and less threatened species.
- Regarding connectivity, we fully agree with the reviewer that this is an important consideration but was unfortunately infeasible due to computational limits. We clarified this above and in the revised text, “Computational limitations associated with optimizing across the large number of planning units (more than 20 million) and features (more than 26,000) prevented use of a contiguity criterion, which would have selected adjacent (contiguous) planning units whenever possible. Consequently, though our results were area efficient, they included prioritized areas that were not contiguous in certain regions and may thus require additional planning for implementation.”
- Regarding the degree of endemism, we set species-specific targets based on each species’ Area of Habitat (AOH) size, and therefore restricted range species (including endemics) should be well represented (e.g. a target of 100% for species with AOH of less than 1,000 km²). We used consistent species targets across all scenarios. (The species targets are explained in the Methods section and we have also added a new table with the species targets to the Supplement (Supplementary Table 3), at the request of one of the other reviewers.)

Reviewer: I have a few other concerns, but I acknowledge that they are minor compared to the ones mentioned earlier. Additionally, considering the extent and resolution of your analyses, it may be challenging to address these concerns effectively. I have intentionally focused on raising concerns that I believe could be helpful in improving your paper. However, I suggest considering the inclusion of a couple of sentences to acknowledge some of the most significant weaknesses of your analysis regarding the input data, extent, and resolution.

- Response: Thank you for this suggestion. Our goal in this paper is to identify countries or regions where conservation attention is urgently needed, and not directly guide conservation interventions on the ground. We have added several key limitations to the Discussion section, including our focus on global optimization and the need for finer-scale information, explaining that our results represent a best case scenario (in which countries coordinate and conservation is optimally allocated), the need for local data on NCP and biodiversity that may not be well represented in globally available data, and explaining that the Development Potential Index may over- or under-estimate development suitability in some regions.
- These are in addition to the limitations already mentioned, including that, while we included a large number of NCP and species, there are other NCP and taxonomic groups that should be added as data becomes available; considerations like evolutionary processes, species traits, intactness of ecosystems, and ecosystem representation should be included in future iterations; more powerful computing resources would enable finer-scale prioritization; and that future work is needed to understand how changes in climate, population, and consumption patterns will affect which ecosystems will be essential for NCP and biodiversity in the future.

I really hope his assessment and point of view could be beneficial in enhancing your paper. Please feel free to contact me if you have any further questions or require clarification. Best regards,

Javier Nori

- Response: we'd like to thank all the reviewers for the many constructive suggestions they made which we believe have strengthened the paper substantially.

References

Allan, J. R., Possingham, H. P., Atkinson, S. C., Waldron, A., Marco, M. D., Adams, V. M.,
Butchart, S. H. M., Venter, O., Maron, M., Williams, B. A., Jones, K. R., Visconti, P.,
Wintle, B. A., Reside, A. E., & Watson, J. E. M. (2019). Conservation attention necessary

across at least 44% of Earth's terrestrial area to safeguard biodiversity. *BioRxiv*, 839977.
<https://doi.org/10.1101/839977>

Brooks, T. M., Mittermeier, R. A., da Fonseca, G. a. B., Gerlach, J., Hoffmann, M., Lamoreux, J. F., Mittermeier, C. G., Pilgrim, J. D., & Rodrigues, A. S. L. (2006). Global Biodiversity Conservation Priorities. *Science*, *313*(5783), 58–61.
<https://doi.org/10.1126/science.1127609>

Brooks, T. M., Pimm, S. L., Akçakaya, H. R., Buchanan, G. M., Butchart, S. H. M., Foden, W., Hilton-Taylor, C., Hoffmann, M., Jenkins, C. N., Joppa, L., Li, B. V., Menon, V., Ocampo-Peñuela, N., & Rondinini, C. (2019). Measuring terrestrial Area of Habitat (AOH) and its utility for the IUCN Red List. *Trends in Ecology & Evolution*, *34*(11), 977–986. <https://doi.org/10.1016/j.tree.2019.06.009>

Cattaneo, A., Nelson, A., & McMenomy, T. (2021). Global mapping of urban–rural catchment areas reveals unequal access to services. *Proceedings of the National Academy of Sciences*, *118*(2), e2011990118. <https://doi.org/10.1073/pnas.2011990118>

Chaplin-Kramer, R., Neugarten, R. A., Sharp, R. P., Collins, P. M., Polasky, S., Hole, D., Schuster, R., Strimas-Mackey, M., Mulligan, M., Brandon, C., Diaz, S., Fluet-Chouinard, E., Gorenflo, L. J., Johnson, J. A., Kennedy, C. M., Keys, P. W., Longley-Wood, K., McIntyre, P. B., Noon, M., ... Watson, R. A. (2022). Mapping the planet's critical natural assets. *Nature Ecology & Evolution*, *7*(1), Article 1. <https://doi.org/10.1038/s41559-022-01934-5>

Díaz, S., Settele, J., Brondízio, E. S., Ngo, H. T., Agard, J., Arneth, A., Balvanera, P., Brauman, K. A., Butchart, S. H. M., Chan, K. M. A., Garibaldi, L. A., Ichii, K., Liu, J., Subramanian, S. M., Midgley, G. F., Miloslavich, P., Molnár, Z., Obura, D., Pfaff, A., ...

- Zayas, C. N. (2019). Pervasive human-driven decline of life on Earth points to the need for transformative change. *Science*, 366(6471). <https://doi.org/10.1126/science.aax3100>
- Esri, Clark Labs, & ESA CCI. (2021). *Converted Lands 2018 to 2050* [dataset]. <https://www.arcgis.com/home/item.html?id=645c280931ac486c92c828eac09e3>
- Hanson, J. O., Rhodes, J. R., Butchart, S. H. M., Buchanan, G. M., Rondinini, C., Ficetola, G. F., & Fuller, R. A. (2020). Global conservation of species' niches. *Nature*, 580(7802), Article 7802. <https://doi.org/10.1038/s41586-020-2138-7>
- Hanson, J. O., Schuster, R., Morrell, N., Strimas-Mackey, M., Watts, M. E., Arcese, P., Bennett, J., & Possingham, H. P. (2020). *prioritizr: Systematic Conservation Prioritization in R* (R package version 5.0.0.0.) [Computer software]. <https://github.com/prioritizr/prioritizr>
- Harris, T., Mulligan, M., & Brummitt, N. (2021). Opportunities and challenges for herbaria in studying the spatial variation in plant functional diversity. *Systematics and Biodiversity*, 19(4), 322–332. <https://doi.org/10.1080/14772000.2021.1887394>
- Hewson, J., Crema, S. C., González-Roglich, M., Tabor, K., & Harvey, C. A. (2019). New 1 km Resolution Datasets of Global and Regional Risks of Tree Cover Loss. *Land*, 8(1), 14. <https://doi.org/10.3390/land8010014>
- Hole, D. G., Collins, P., Tesfaw, A., Barrera, L., Mascia, M. B., & Turner, W. R. (2022). Make nature's role visible to achieve the SDGs. *Global Sustainability*, 5, e8. <https://doi.org/10.1017/sus.2022.5>
- Joppa, L. N., & Pfaff, A. (2009). High and Far: Biases in the Location of Protected Areas. *PLOS ONE*, 4(12), e8273. <https://doi.org/10.1371/journal.pone.0008273>
- Jung, M., Arnell, A., de Lamo, X., García-Rangel, S., Lewis, M., Mark, J., Merow, C., Miles, L., Ondo, I., Pironon, S., Ravilious, C., Rivers, M., Schepaschenko, D., Tallowin, O., van

- Soesbergen, A., Govaerts, R., Boyle, B. L., Enquist, B. J., Feng, X., ... Visconti, P. (2021). Areas of global importance for conserving terrestrial biodiversity, carbon and water. *Nature Ecology & Evolution*, 5(11), 1499–1509. <https://doi.org/10.1038/s41559-021-01528-7>
- Myers, N. (2003). Biodiversity Hotspots Revisited. *BioScience*, 53(10), 916–917. [https://doi.org/10.1641/0006-3568\(2003\)053\[0916:BHR\]2.0.CO;2](https://doi.org/10.1641/0006-3568(2003)053[0916:BHR]2.0.CO;2)
- Riggio, J., Baillie, J. E. M., Brumby, S., Ellis, E., Kennedy, C. M., Oakleaf, J. R., Tait, A., Tepe, T., Theobald, D. M., Venter, O., Watson, J. E. M., & Jacobson, A. P. (2020). Global human influence maps reveal clear opportunities in conserving Earth's remaining intact terrestrial ecosystems. *Global Change Biology*, 26(8), 4344–4356. <https://doi.org/10.1111/gcb.15109>
- Rodrigues, A. S. L., Akçakaya, H. R., Andelman, S. J., Bakarr, M. I., Boitani, L., Brooks, T. M., Chanson, J. S., Fishpool, L. D. C., Da Fonseca, G. A. B., Gaston, K. J., Hoffmann, M., Marquet, P. A., Pilgrim, J. D., Pressey, R. L., Schipper, J., Sechrest, W., Stuart, S. N., Underhill, L. G., Waller, R. W., ... Yan, X. (2004). Global gap analysis: Priority regions for expanding the global protected-area network. *BioScience*, 54(12), 1092–1100. [https://doi.org/10.1641/0006-3568\(2004\)054\[1092:GGAPRF\]2.0.CO;2](https://doi.org/10.1641/0006-3568(2004)054[1092:GGAPRF]2.0.CO;2)
- Rosenberg, K. V., Dokter, A. M., Blancher, P. J., Sauer, J. R., Smith, A. C., Smith, P. A., Stanton, J. C., Panjabi, A., Helft, L., Parr, M., & Marra, P. P. (2019). Decline of the North American avifauna. *Science*, 366(6461), 120–124. <https://doi.org/10.1126/science.aaw1313>
- Sala, E., Mayorga, J., Bradley, D., Cabral, R. B., Atwood, T. B., Auber, A., Cheung, W., Costello, C., Ferretti, F., Friedlander, A. M., Gaines, S. D., Garilao, C., Goodell, W., Halpern, B. S.,

- Hinson, A., Kaschner, K., Kesner-Reyes, K., Leprieur, F., McGowan, J., ... Lubchenco, J. (2021). Protecting the global ocean for biodiversity, food and climate. *Nature*, *592*(7854), 397–402. <https://doi.org/10.1038/s41586-021-03371-z>
- Sayre, R., Karagulle, D., Frye, C., Boucher, T., Wolff, N. H., Breyer, S., Wright, D., Martin, M., Butler, K., Van Graafeiland, K., Touval, J., Sotomayor, L., McGowan, J., Game, E. T., & Possingham, H. (2020). An assessment of the representation of ecosystems in global protected areas using new maps of World Climate Regions and World Ecosystems. *Global Ecology and Conservation*, *21*, e00860. <https://doi.org/10.1016/j.gecco.2019.e00860>
- Schuster, R., Wilson, S., Rodewald, A. D., Arcese, P., Fink, D., Auer, T., & Bennett, J. R. (2019). Optimizing the conservation of migratory species over their full annual cycle. *Nature Communications*, *10*(1), Article 1. <https://doi.org/10.1038/s41467-019-09723-8>
- Shen, X., Liu, M., Hanson, J. O., Wang, J., Locke, H., Watson, J. E. M., Ellis, E. C., Li, S., & Ma, K. (2023). Countries' differentiated responsibilities to fulfill area-based conservation targets of the Kunming-Montreal Global Biodiversity Framework. *One Earth*, *6*(5), 548–559. <https://doi.org/10.1016/j.oneear.2023.04.007>
- Tickner, D., Opperman, J. J., Abell, R., Acreman, M., Arthington, A. H., Bunn, S. E., Cooke, S. J., Dalton, J., Darwall, W., Edwards, G., Harrison, I., Hughes, K., Jones, T., Leclère, D., Lynch, A. J., Leonard, P., McClain, M. E., Muruven, D., Olden, J. D., ... Young, L. (2020). Bending the Curve of Global Freshwater Biodiversity Loss: An Emergency Recovery Plan. *BioScience*, *70*(4), 330–342. <https://doi.org/10.1093/biosci/biaa002>

UN CBD. (2022). *Kunming-Montreal Global biodiversity framework: Draft decision submitted by the President* (CBD/COP/15/L.25). UN Convention on Biological Diversity.

<https://www.cbd.int/doc/c/e6d3/cd1d/daf663719a03902a9b116c34/cop-15-l-25-en.pdf>

U.S. Department of Interior (DOI). (2021). *Conserving and restoring America the Beautiful*.

DOI, U.S. Department of Agriculture, U.S. Department of Commerce, and Council on Environmental Quality. <https://www.doi.gov/sites/doi.gov/files/report-conserving-and-restoring-america-the-beautiful-2021.pdf>

Weiss, D. J., Nelson, A., Gibson, H. S., Temperley, W., Peedell, S., Lieber, A., Hancher, M., Poyart, E., Belchior, S., Fullman, N., Mappin, B., Dalrymple, U., Rozier, J., Lucas, T. C. D., Howes, R. E., Tusting, L. S., Kang, S. Y., Cameron, E., Bisanzio, D., ... Gething, P. W. (2018). A global map of travel time to cities to assess inequalities in accessibility in 2015. *Nature*, 553(7688), Article 7688. <https://doi.org/10.1038/nature25181>

Xu, H., Cao, Y., Yu, D., Cao, M., He, Y., Gill, M., & Pereira, H. M. (2021). Ensuring effective implementation of the post-2020 global biodiversity targets. *Nature Ecology & Evolution*, 1–8. <https://doi.org/10.1038/s41559-020-01375-y>

REVIEWERS' COMMENTS

Reviewer #2 (Remarks to the Author):

I appreciate all the time and thought that went into the changes and response to my comments. I still do not understand how the authors recommend interpreting and acting on their core finding (90% of NCP being provided by 44% of land if you ignore existing protected areas, minimum patch size and logistical constraints, equity across nations). However, as that's a question about the framing rather than the analysis it should not be a barrier to the paper being published if the authors are satisfied with their message.

I have provided a handful of additional minor comments to try and make it more obvious to the reader what the findings do and don't mean.

Line 31: "approximately half" in the abstract is a good solution to the 44% vs 49% issue. Nicely done.

Line 75: I recommend adding the following text to the sentence ending in 'species representation targets' to ensure the reader doesn't misinterpret this result: "(independent of protection status or constraints on equity)"

Line 98: I recommend explicitly stating something like 'Natural assets for NCP are distributed very unevenly across countries, with critical areas including...' It isn't until the discussion that the reader discovers that these results include some countries having over half of their land areas set aside for conservation (which is striking as it's highly unlikely those countries would agree to that even if paid). It's an important result to caveat how the high level numbers are interpreted.

Line 182: I recommend adding one more sentence like "The logistical challenges of small non-contiguous priority areas, and the likelihood that not all areas would be effectively conserved also likely imply more land area would be needed to achieve species and NCP targets.

Line 184: What kinds of 'broad action' do the authors have in mind for applying their results? Providing funding and other support to countries with high levels of priority areas could be one, but adding 2-3 clear examples would help a lot.

Lines 200-202: This seems in conflict with recommending that more than half the land of some countries be set aside for conservation. Can this be explicitly stated as a trade-off or

concern? Essentially - since the results are so optimized, if locals don't agree w/ the results (as often happens when locals are asked to react to a map outsiders made) it could substantially increase the amount of land needed.

Lines 212-214: I continue to struggle with a meaningful way to interpret this finding. The authors seem to imply that expanding protected areas is an insufficient approach, which again would lead the reader to think that they're recommending some protected areas be unprotected (or swapped with higher priority lands?). Whatever the authors think should happen could be more clearly stated. The rebuttal said it wasn't unprotecting existing protected areas, but if it's not that nor locking them in and adding new ones, what is left? Given all the research showing existing protected areas perform poorly for biodiversity, threat, and representation, proposing some kind of swap could be a good idea! But whatever the idea is, please help the reader to see it.

Line 244: Funding alone would likely not be enough - many countries simply will not want to set aside large portions of their land (especially more than half) even if paid for. In many areas, farmers and ranchers prefer to keep farming and ranching than to be paid to do something else. This should be acknowledged (could be super brief).

Lines 398-400: This sentence is highly misleading. If you want to understand how much land area is actually required to meet a target (and thus what target should be set), logistical concerns like minimum patch size (or perimeter:area ratio or whatever) are very important! It would be both politically untenable and logistically impractical to suggest a scattered network of 10 km² conservation areas across a country. The computational limitation makes perfect sense and is adequate justification, but the sentence from lines 398-400 should be deleted. I don't think it's true that having no minimum area constraint is the best way to determine how much land area is actually needed to hit targets. Just stick with it as a computational limitation.

Apologies for having such extensive comments - I realize I am truly "reviewer #2" in this instance. I think the analysis is interesting and with the relatively small changes to framing I recommend I think the paper will stimulate a lot of good discussions in conservation.

Reviewer #3 (Remarks to the Author):

Dear Colleagues,

First and foremost, I would like to apologize for the delay in providing my assessment, which, I believe, was mainly due to my own schedule constraints.

Having said that, I want to express my appreciation for the tremendous effort put into the revision process. I have thoroughly reviewed the responses and the revised version of the paper, and I must congratulate you on your excellent work. The responses are not only clear and well-developed but also, most importantly, the changes made go beyond mere cosmetic adjustments. They included new information, analyses, and, in my view, significant enhancements to the text. Overall, I am entirely satisfied with the outcome of the revision.

Regarding my second point, I still maintain my belief that the inclusion of a bit more complex analysis as a biodiversity proxy could have been advantageous. Nevertheless, I am also fully content with the response and the justification provided by the authors. I believe that the paper presents compelling data, analyses, and findings.

In conclusion, in my capacity as a reviewer, I wish to convey to the editor my positive overall perspective on this document.

Congratulations to all

regards,

Javier

REVIEWERS' COMMENTS

Reviewer #2 (Remarks to the Author):

I appreciate all the time and thought that went into the changes and response to my comments. I still do not understand how the authors recommend interpreting and acting on their core finding (90% of NCP being provided by 44% of land if you ignore existing protected areas, minimum patch size and logistical constraints, equity across nations). However, as that's a question about the framing rather than the analysis it should not be a barrier to the paper being published if the authors are satisfied with their message.

- Response: We apologize that our recommendations are still unclear. Our intention is to identify areas requiring conservation attention, where “conservation” could be achieved through numerous potential strategies, including but not limited to protected areas and OECMs, and even sustainable management if such management is consistent with conservation outcomes. The Kunming-Montreal Global Biodiversity Framework Target 3, for example, states that “by 2030 at least 30 per cent of terrestrial, inland water, and of coastal and marine areas, especially areas of particular importance for biodiversity and ecosystem functions and services, are effectively conserved and managed” and mentions protected areas, OECM, indigenous territories, and sustainable use that is fully consistent with conservation outcomes”. Also, because our focus here is on priorities for nature’s contributions to people (NCP), not just biodiversity, and because certain NCP (timber, fuelwood, grazing, access to nature) require ongoing human access and use (sustainable management) in order to continue to benefit people, strict protection may not be appropriate in many cases. We have clarified this in the Introduction:
 - “In an attempt to prevent further losses of biodiversity and NCP, nearly 200 nations have recently committed to **effectively conserving and managing** 30% of lands and waters by 2030 under the Kunming-Montreal Global Biodiversity Framework⁴ and similar national targets (e.g. “America the Beautiful”⁵).”
- And we have also emphasized and clarified this in the Discussion:
 - “While our maps identify areas in urgent need of conservation attention, they are not intended to define priorities for strict protection. Strictly protected areas preclude activities such as grazing or timber harvesting which are essential to the provision of certain NCP. Furthermore, the current PA and OECM networks are disproportionately located in remote areas with relatively low threat²⁹, and do not represent important areas for NCP particularly well (Supplementary Fig. 2), as NCP tend to be concentrated in areas with natural and semi-natural habitat in proximity to human populations. **Other conservation measures, including OECMs, strengthening Indigenous and local land tenure, Payments for Ecosystem Services (PES), and sustainable management will be essential for conserving NCP and biodiversity outside of the current system of protected areas. For example, areas that are providing high levels of water quality, flood regulation, and timber production could be targeted for PES, certification, or other mechanisms. Areas required to achieve species targets that also contain vulnerable carbon could be candidates for Indigenous, local, or government protection; but methods other than**

protection can also be effective at maintaining biodiversity and carbon stocks.”

- Later in the Discussion we had also tried to explain:
 - “Certain forms of development, if appropriately located and carefully designed, may be compatible with the ongoing provision of NCP and biodiversity conservation. Examples include water-sensitive urban design that enhances biodiversity, such as green roofs and rain gardens³² and solar energy farms that can double as livestock enclosures³³, enhance crop production³⁴ or provide habitat for pollinators and other ecosystem services³⁵.”

I have provided a handful of additional minor comments to try and make it more obvious to the reader what the findings do and don't mean.

Line 31: "approximately half" in the abstract is a good solution to the 44% vs 49% issue. Nicely done.

- Response: We thank the reviewer for these additional suggestions and have incorporated all of them.

Line 75: I recommend adding the following text to the sentence ending in 'species representation targets' to ensure the reader doesn't misinterpret this result: "(independent of protection status or constraints on equity)"

- Response: Thank you for the suggestion, we added “independent of protection status” to the sentence, and added a separate sentence: “We did not set area constraints by country.” (We thought “constraints on equity” might be hard for a typical reader to interpret.)

Line 98: I recommend explicitly stating something like 'Natural assets for NCP are distributed very unevenly across countries, with critical areas including...' It isn't until the discussion that the reader discovers that these results include some countries having over half of their land areas set aside for conservation (which is striking as it's highly unlikely those countries would agree to that even if paid). It's an important result to caveat how the high level numbers are interpreted.

- Done, thank you.

Line 182: I recommend adding one more sentence like "The logistical challenges of small non-contiguous priority areas, and the likelihood that not all areas would be effectively conserved also likely imply more land area would be needed to achieve species and NCP targets.

- Done.

Line 184: What kinds of 'broad action' do the authors have in mind for applying their results? Providing funding and other support to countries with high levels of priority areas could be one, but adding 2-3 clear examples would help a lot.

- Response: We have revised this paragraph (new text in bold):
- “By providing consistent and comparable data across countries, global maps can facilitate the establishment of international targets and highlight where broad action and investment may be most impactful²⁶. Our work builds upon previous efforts that focused on national-scale NCP priorities that accrue benefits at local- to regional-scales¹ by including areas required for conserving vulnerable ecosystem carbon stocks (a global benefit) as well as biodiversity. Our aim in this paper was to identify global-scale priorities to support processes such as the Global Biodiversity Framework, **and to inform funding priorities of actors with a worldwide remit. Within priority areas, national and sub-national planning also can benefit from understanding the global significance of local conservation efforts. Global priorities also can support efforts of less wealthy nations to secure resources to achieve shared global targets.**”

Lines 200-202: This seems in conflict with recommending that more than half the land of some countries be set aside for conservation. Can this be explicitly stated as a trade-off or concern? Essentially - since the results are so optimized, if locals don't agree w/ the results (as often happens when locals are asked to react to a map outsiders made) it could substantially increase the amount of land needed.

- Response: We agree that the amount of land needed could be substantially increased, as it is unlikely that conservation action and investment (including sustainable management, OECMs, or other appropriate mechanisms) will always be targeted to optimal locations. We have added this sentence:
- “Also, given the many barriers to optimally targeting conservation action and investments, our estimates of the area required to achieve targets are likely to be conservative.”
- We do not recommend that “more than half the land of some countries be set aside for conservation”, which we tried to clarify above, but rather that such areas require more attention.

Lines 212-214: I continue to struggle with a meaningful way to interpret this finding. The authors seem to imply that expanding protected areas is an insufficient approach, which again would lead the reader to think that they're recommending some protected areas be unprotected (or swapped with higher priority lands?). Whatever the authors think should happen could be more clearly stated. The rebuttal said it wasn't unprotecting existing protected areas, but if it's not that nor locking them in and adding new ones, what is left? Given all the research showing existing protected areas perform poorly for biodiversity, threat, and representation, proposing some kind of swap could be a good idea! But whatever the idea is, please help the reader to see it.

- Response: In terms of “what is left?”, we see a multitude of potential approaches. The appropriate approach for a particular location should be tailored to the specific conservation values (biodiversity, carbon, freshwater provision, grazing/fodder production, recreation, etc.) and to the local context. That said, we agree a swap could be appropriate in certain cases, where existing protected areas are not optimally located and/or resources could be more effectively allocated to other areas. However, given the

limitations of our analyses (taxonomic limitations, limited set of NCPs for which data is currently available, etc.) we do not recommend de-gazetting areas on the basis of our maps alone.

Line 244: Funding alone would likely not be enough - many countries simply will not want to set aside large portions of their land (especially more than half) even if paid for. In many areas, farmers and ranchers prefer to keep farming and ranching than to be paid to do something else. This should be acknowledged (could be super brief).

- Response: Agreed. We hope the additional explanation and revisions provided above (that we are not recommending “setting aside” large portions of land) is helpful. Our maps of priorities for NCP include pollination for agriculture and fodder production for grazing, and include both natural and semi-natural (e.g. grazed pasture, commercial forestry) landcover classes. In such areas, the goal would be to maintain sustainable flows of NCP while also conserving biodiversity.

Lines 398-400: This sentence is highly misleading. If you want to understand how much land area is actually required to meet a target (and thus what target should be set), logistical concerns like minimum patch size (or perimeter:area ratio or whatever) are very important! It would be both politically untenable and logistically impractical to suggest a scattered network of 10 km² conservation areas across a country. The computational limitation makes perfect sense and is adequate justification, but the sentence from lines 398-400 should be deleted. I don't think it's true that having no minimum area constraint is the best way to determine how much land area is actually needed to hit targets. Just stick with it as a computational limitation.

- Response: We have deleted the sentence.

Apologies for having such extensive comments - I realize I am truly "reviewer #2" in this instance. I think the analysis is interesting and with the relatively small changes to framing I recommend I think the paper will stimulate a lot of good discussions in conservation.

- Response: We thank the reviewer for the generous amount of time and careful attention provided in both reviews, which has helped us clarify and improve the paper significantly.

Reviewer #3 (Remarks to the Author):

Dear Colleagues,

First and foremost, I would like to apologize for the delay in providing my assessment, which, I believe, was mainly due to my own schedule constraints.

Having said that, I want to express my appreciation for the tremendous effort put into the revision process. I have thoroughly reviewed the responses and the revised version of the

paper, and I must congratulate you on your excellent work. The responses are not only clear and well-developed but also, most importantly, the changes made go beyond mere cosmetic adjustments. They included new information, analyses, and, in my view, significant enhancements to the text. Overall, I am entirely satisfied with the outcome of the revision.

Regarding my second point, I still maintain my belief that the inclusion of a bit more complex analysis as a biodiversity proxy could have been advantageous. Nevertheless, I am also fully content with the response and the justification provided by the authors. I believe that the paper presents compelling data, analyses, and findings.

In conclusion, in my capacity as a reviewer, I wish to convey to the editor my positive overall perspective on this document.

Congratulations to all

regards,

Javier

- Response: We thank the reviewer for the very kind words.